# Network curvature as a hallmark of brain structural connectivity

Hamza Farooq[1]*, Yongxin Chen[2], Tryphon T. Georgiou ![ORCID] [3], Allen Tannenbaum[4] & Christophe Lenglet ![ORCID] [5]

Although brain functionality is often remarkably robust to lesions and other insults, it may be fragile when these take place in specific locations. Previous attempts to quantify robustness and fragility sought to understand how the functional connectivity of brain networks is affected by structural changes, using either model-based predictions or empirical studies of the effects of lesions. We advance a geometric viewpoint relying on a notion of network curvature, the so-called Ollivier-Ricci curvature. This approach has been proposed to assess financial market robustness and to differentiate biological networks of cancer cells from healthy ones. Here, we apply curvature-based measures to brain structural networks to identify robust and fragile brain regions in healthy subjects. We show that curvature can also be used to track changes in brain connectivity related to age and autism spectrum disorder (ASD), and we obtain results that are in agreement with previous MRI studies.

[1] Department of Electrical and Computer Engineering, University of Minnesota, Minneapolis, MN, USA. [2] School of Aerospace Engineering, Georgia Institute of Technology, Atlanta, GA, USA. [3] Department of Mechanical and Aerospace Engineering, University of California, Irvine, CA, USA. [4] Departments of Computer Science and Applied Mathematics & Statistics, Stony Brook University, Stony Brook, NY, USA. [5] Center for Magnetic Resonance Research, University of Minnesota, Minneapolis, MN, USA. *email: faroo014@umn.edu

This paper describes a novel geometric network-theoretic approach to study brain structural connectivity. Data for our study is provided by imaging techniques, such as diffusion MRI (dMRI), that are used to map the structural connectivity between different brain regions[1–3]. At a macroscopic scale, brain regions are delineated and perceived as nodes of a network with edges describing connectivity (structural or functional) between them. The overall structure of the brain, at that scale, may be mathematically represented as a graph[4,5]. Depending on the method used to identify edges and determine their relative strengths, brain networks can be divided into three types[4–6]: (i) structural networks with edge weights based on the strength of anatomical links between nodes; (ii) functional networks in which the edges are given by statistical inter-dependence of signals at each node; and (iii) functional networks whose edges are based on the causal influence of nodes. The method employed to spatially parcellate the brain and consequently construct nodes will also affect the network parameters[7,8]. A salient feature of our approach is that it relies on a certain inherently persistent characteristic of nodes, their potential role as hubs within the overall structure, and thereby reflects on a newly introduced notion of robustness of the network as a whole.

In general, robustness of a (brain) network is defined as the "degree to which the topological properties of a network are resilient to lesions such as the removal of nodes or edges[9]." In particular, robustness quantifies to what extent the brain can withstand damage from, or be affected by, lesions arising, e.g. from tumors, trauma, or stroke. Reduced robustness not only suggests potential for dysfunction due to the lesion, but may also point to candidate target locations for treatment.

Brain resilience has been studied previously by considering the effects of deleting nodes or edges from structural and functional networks, both computationally and empirically (see ref. [10] for a comprehensive review). Brain robustness studies can broadly be divided into two categories. In the first category, one attempts to predict the lesion effects by computational models, i.e., virtually removing or modifying nodes and edges of the structural connectivity matrix and applying computational models to predict functional connectivity changes[11–13]. Subsequently, the predicted functional connectivity matrix can be compared with the empirical one and the lesion effects can be quantified using various graph measures. In the second category, one employs the empirical effects from brain lesions due to injury or disease. Studies using this approach focus on examining brain networks of patients with, e.g. traumatic brain injury (TBI), stroke or tumors, and quantify the effect of lesion location on the brain[14,15], by comparison with data from age-matched and gender-matched healthy control subjects. Regardless of the approach, the structure-to-function network relation is utilized to predict the amount of damage which the brain can withstand due to lesions in a given location.

We apply the geometric notion of graph curvature to brain structural networks, and leverage this novel concept to analyze brain robustness. Previous studies have shown that network curvature can be used to differentiate cancer from normal tissue using gene co-expression networks[16], and to indicate market fragility in economic or financial networks[17]. It is important to note that, since network robustness can be viewed as the rate function at which a network returns to its original state after a perturbation, it has a positive correlation with entropy[18]. Consequently, network robustness and curvature are positively correlated through entropy[19]. A detailed mathematical characterization of the concept of graph curvature is provided in the "Methods" section.

In this paper, we introduce the concept of graph curvature for studying brain structural connectivity networks. More specifically,

we use the Ricci curvature and its contraction, the scalar curvature, on brain networks so as to assign curvature at each individual node. Thus, by introducing such a notion of nodal measure, we make two distinct contributions to brain structural connectivity analysis: First, we identify areas of the brain that significantly contribute to the overall brain robustness, and hence we identify "important" nodes in brain networks. Previous studies have shown that hub nodes are critical for brain networks, but identifying such nodes is not straightforward. Node measures such as degree or strength do not identify all the hub nodes, and typically a combination of those measures, with centrality measures, is required[10,20]. We show that node curvature not only corroborates findings based on strength and centrality measures, but additionally finds other key areas (e.g., inferior-frontal gyrus, middle-frontal gyrus, and inferior-temporal gyrus), which are not identified by any other node measure, and are important parts of the brain network. Second, by looking at differences in node curvature, one can identify brain areas with changes due to age, or abnormal neurodevelopment disorders such as autism spectrum disorders (ASD). In particular, we show that node curvature uniquely enables the identification of certain brain areas, with significantly affected structural connectivity in people with ASD.

## Results

**Curvature as a hallmark of brain areas robustness.** Individual node curvature (defined in "Methods" section, Eq. (11)) of brain areas contributes to the overall (average) curvature of the brain network. This measure not only helps identify alterations in the network, but also can help identify key (i.e. important) parts of the brain structural network. As explained in the "Methods" "section, Eq. (12), curvature is directly correlated with network robustness. Therefore, nodes with higher curvature contribute more to the overall structural robustness of the network.

To demonstrate this, we performed experiments using two different diffusion spectrum imaging (DSI) datasets: First, the DSI data for five participants, as presented in refs. [11,20], was considered, to enable comparison of our results with previous studies. High-resolution connectivity matrices ($998 \times 998$) were obtained from the USC Multimodal Connectivity Database[21]. Second, the DSI data for 33 participants from the MGH-USC HCP Consortium was also employed, and lower resolution connectivity matrices ($116 \times 116$) were generated (as described in the "Methods" section, DSI Datasets from the MGH-USC HCP Consortium)[22].

As previously described, we emphasize here that comparing properties across brain networks with different resolutions (i.e. number of nodes) should be done only with great care[7], as brain network properties can differ significantly with nodal parcellations[7,8]. Nonetheless, it is worth studying curvature as a measure which may provide information across different network resolutions: high-resolution parcellations, also known as dense connectomes[23] will ultimately provide greater insights into the structure of brain networks, while lower resolution parcellations are more easily manageable, since they requires less computational resources.

First, we present results based on the high-resolution connectivity matrices[21] in Fig. 1. Here, we show the top 25% of nodes with the highest node curvature, strength and betweenness centrality, appearing consistently across the five participants. Figures in panels b and c of Fig. 1 follow the same convention as Figs. 2C and 7A of Hagmann et al.[20], respectively, and are presented here for comparison purposes. The details of the areas identified by all three measures can be found in Supplementary Note 1. In the previous study[20], using several network analysis methods, eight anatomical regions were identified as belonging to

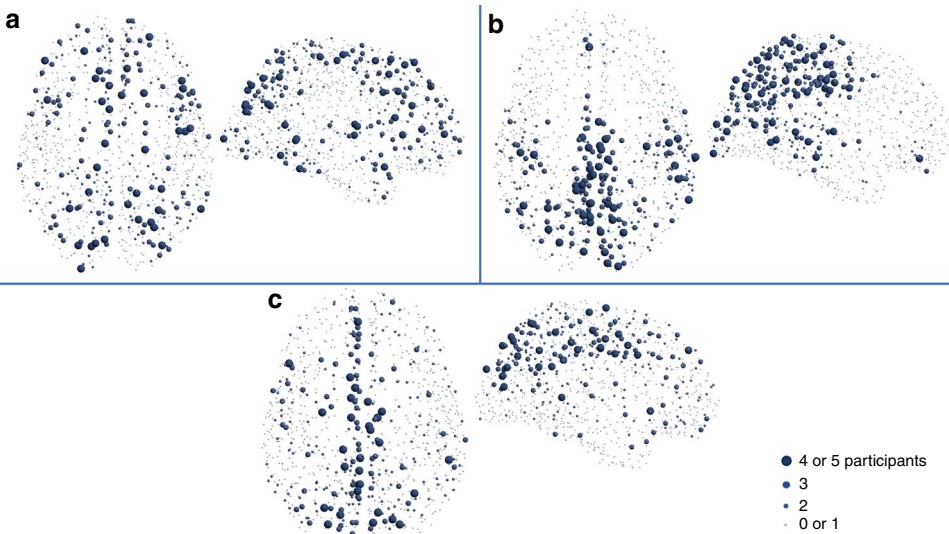

**Fig. 1** Brain areas with the highest nodal measures appearing consistently across the participants, using high-resolution connectivity matrices (998 × 998) from Hagmann et al. [20]. The top 25% nodes are shown for curvature (**a**), strength (**b**), and betweenness centrality (**c**). For instance, the largest spheres indicate nodes with high values in 4 or 5 out of the five subjects

the so-called structural core network of the human brain. These regions include the posterior cingulate cortex, precuneus, cuneus, paracentral lobule, isthmus of the cingulate, banks of the superior temporal sulcus, and inferior/superior parietal cortex, all of them in both hemispheres.

Additionally, Alstott et al.[11] showed that lesions in the temporo-parietal junction, cortical midline and frontal cortex have the most extensive effects on brain functionality. Also, we note that the medial prefrontal cortex forms part of the default mode network of the human brain[24]. Panel a of Fig. 1 shows that curvature identifies areas in the inferior-frontal gyrus, middle-frontal gyrus and inferior-temporal gyrus, consistent with[11,20], and thus providing very interesting information based on network structure, which is not captured by strength or betweenness centrality.

Second, following the same organization as Fig. 1, Fig. 2 shows results for the lower resolution matrices generated from the MGH-USC HCP Consortium datasets. As expected, distinct areas are identified with all three measures (since cortical parcellation is different from the one used in Fig. 1)[7]. We should also note that the high-resolution data did not include the cerebellum. Nodes with high strength and betweenness centrality are found more towards the frontal, precentral, superior parietal areas, and in the cerebellum. Once again, curvature supplements the information provided by other measures and identifies areas in the inferior-frontal gyrus and transverse temporal gyrus (Heschl's gyrus) in both hemispheres, where lesions are known to induce pronounced effects in loss of brain functionality[11] (see the list of areas in Supplementary Note 2).

**Curvature changes in different age groups**. We used datasets from the WU-Minn HCP Consortium Lifespan Pilot Project to study structural changes in brain networks related to aging in groups of independent participants. Details about the data and construction of connectivity matrices, using a set of 333 areas[25], are given in the "Methods" section (HARDI datasets from the WU-Minn HCP Consortium Lifespan Pilot Project https://www.humanconnectome.org/study-hcp-lifespan-pilot). In Fig. 3, we show areas with statistically significant differences in nodal measures, related to aging between the Lifespan group 2 (age 8–9) and group 6 (age 65–75). For the results shown, family-wise error

rate was controlled using the Holm–Sidak[26] method, details given in the "Methods" section (family-wise error correction). Results are also shown in tabular form in Supplementary Note 3.

Node measures such as strength, betweenness centrality, and clustering coefficient can provide useful information about areas involved in aging. That is, consistent with the previous studies, these measures collectively find significant bilateral differences in the visual areas[27,28], dorsal parietal lobe[29], cingulo-opercular network regions[30], and temporal areas[31,32]. Focusing on information uniquely provided by the node curvature, we see that the measure identifies significant structural changes in the areas known to change with age from previous literature, while not identified by other measures like the cingulo-parietal network[33], right visual cortex[34,35], and lateral occipital areas[36]. Thus, node curvature provides information complementary to other node measures revealing structural changes due to age with more details.

**Curvature differences in ASD**. The aim of this analysis is to test whether various measures of node importance or robustness (curvature, strength, centrality, and clustering) can detect differences in structural connectivity between individuals with ASD and typically developing (TD) subjects. We utilized diffusion tensor imaging (DTI) data from San Diego State University (SDSU) and Trinity Center for Health Sciences (TC) available from the Autism Brain Imaging Data Exchange II (ABIDE-II)[37] http://fcon_1000.projects.nitrc.org/indi/abide/abide_II.html). Details about the data are given in the "Methods" section (DTI datasets from ABIDE-II[37]). We used 29 ASD and 24 TD subjects from SDSU data, and 20 ASD and 20 TD data from TC. DTI connectivity matrices capturing the brain structural connectivity of each participant were generated using a set of 333 areas[25] in MNI space (see in the "Methods" section, section "Generation of connectivity matrices", ABIDE-II Datasets)

Figure 4 shows areas with statistically significant differences between the ASD and TD groups, identified by node measures using both datasets. For the results shown, family-wise error rate (e.g. type I error) was controlled using the Holm–Sidak[26] method, and details are provided in the "Methods" section (Family-wise error correction). Results are also shown in tabular form in Supplementary Note 4.

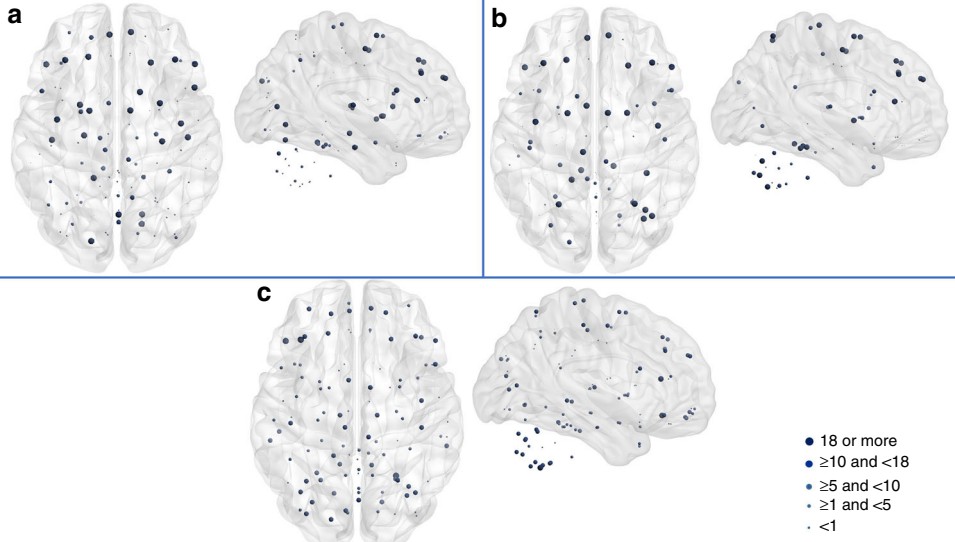

**Fig. 2** Brain areas with the highest nodal measures appearing consistently across the participants, using lower resolution connectivity matrices (116 × 116) generated using the AAL atlas and MGH-USC DSI datasets. The top 25% nodes are shown for curvature (**a**), strength (**b**), and betweenness centrality (**c**). Here, the largest spheres indicate nodes with high values in 18 out of the 33 subjects

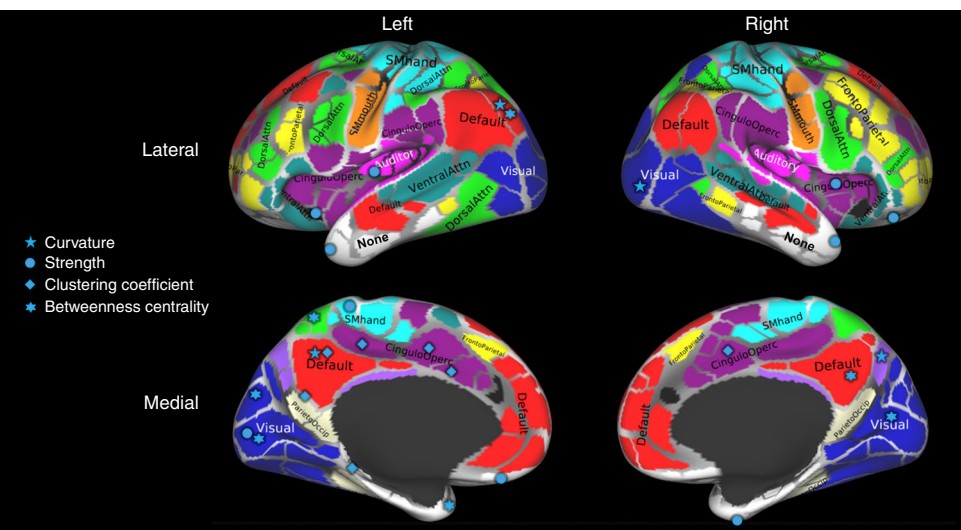

**Fig. 3** Nodes with statistically significant changes (corrected for multiple comparisons using the Holm–Sidak method) in structural connectivity due to age. Brain parcellation with 333 cortical areas was done using the Gordon atlas[25] and labeled using the Brain Analysis Library of Spatial maps and Atlases database https://balsa.wustl.edu/WK71. Adapted from Fig. 10 of Supplementary data from Gordon et al.[25]

Left-lateralized patterns of abnormalities in the brain microstructure and connectivity are known from previous studies[38,39]. The same pattern can be seen in Fig. 4 as collectively, node measures find more differences in the left hemisphere. Mostly, affected nodes were identified in the temporal lobe, visual and auditory cortices, default mode areas, and the somatomotor hand areas of the left hemisphere. While in the right hemisphere, changes were found in the dorsal attention network areas, visual, somatomotor hand, and retrosplenial temporal areas.

The left temporal pole is related to semantic memory[40] and previous studies[41,42] present evidence of changes of the temporal pole in ASD. Node curvature, strength, and betweenness centrality all identify significant changes in this area. However, the right temporal lobe is only identified by node curvature, in agreement with a recent study[43] identifying microstructural changes in that area, due to ASD and relating the changes with

communication impairment. Also, the right temporal pole is associated with emotions and socially relevant memories[40], which are affected in ASD.

Studies have shown a difference in visual perception in patients with ASD, compared to TD. For example, patients with ASD perform better in detecting visual targets in a large field of view, and are also more detail oriented[44–46]. Structural connectivity changes due to ASD in the right occipital lobe (right visual areas) were also reported in voxel-based morphometry study[47]. Here in our analysis, node curvature finds changes in both hemispheres, while other node measures find changes only in the left visual area. Curvature also identifies dorsal attention network areas which, from previous literature, are known to be affected in ASD[48]. To summarize, curvature may provide new information about brain connectivity patterns in ASD, which is complementary to previous studies using morphometric and weighed-graph

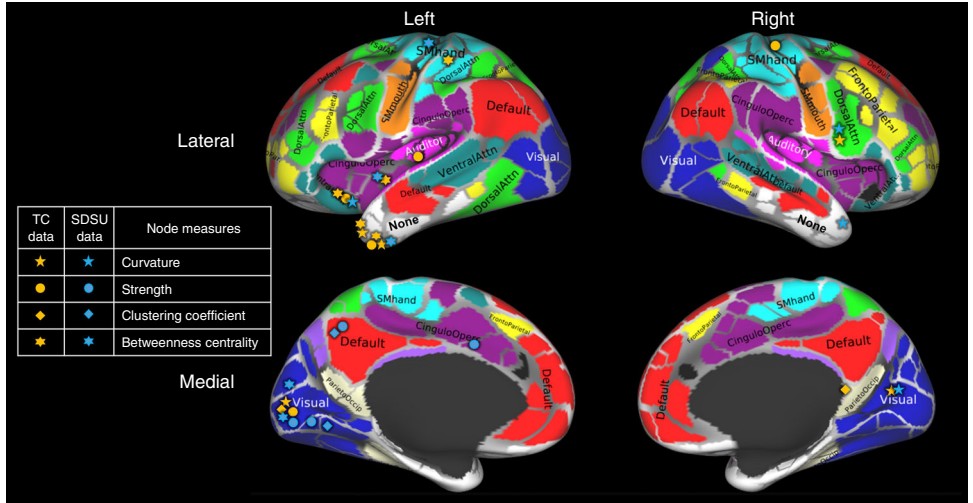

**Fig. 4** Nodes with statistically significant differences (corrected for multiple comparisons using the Holm–Sidak method) in structural connectivity between individuals with ASD and TD subjects. Nodes identified using either the San Diego State University (SDSU) or the Trinity Center for Health Sciences (TC) data are shown in different colors (blue and orange, respectively). Brain parcellation with 333 cortical areas was done using the Gordon atlas[25] and labeled using the Brain Analysis Library of Spatial maps and Atlases database https://balsa.wustl.edu/WK71. Adapted from Fig. 10 of Supplementary data from Gordon et al.[25]

node measures[49,50]. Therefore, curvature may provide new information about brain connectivity patterns in ASD, which is complementary to previously shown morphometric alterations of specific brain areas.

In order to gain further insights into structural connectivity disruptions in ASD, we performed a univariate analysis to study the relationships between nodal measures with significant differences related to ASD, and affected phenotypic measures from the ABIDE-II database[37]. Curvature of the right temporal lobe and Social Responsiveness Scale[51] (SRS) sub-factor Motivation (both Raw and converted T-scores), and Repetitive Behavior Scale-Revised (RBSR), are found to be negatively correlated. The left orbito-frontal cortex curvature was also uniquely identified to correlate positively with the Autism Diagnostic Observation Schedule (ADOS-2) Restricted and Repetitive behavior scale. Additionally, curvature of the anterior division of the temporal fusiform cortex positively correlates with several Child Behavior Checklist (CBCL) scores (e.g. Attention, Aggressive behavior) and RBSR sub-factor Self-injurious behavior. This is in line with prior studies[52,53] and supplement the information provided by other node measures. Correlation plots and additional details are provided in Supplementary Note 5.

**Brain networks properties and robustness characterization**. In this section, we discuss how brain network properties (robustness in particular) can be assessed using graph measures. We have shown that curvature can detect brain areas that are critical, although not identified by other measures, as well as areas related to age or abnormal neurodevelopment in ASD. Here, we further examine how curvature might provide a novel method to study brain robustness, complementary to other graph measures.

In order to quantify the robustness of a given node in a brain network, the effect of node(s) deletion on graph measures can be considered[11]. Based on nodal "importance" measures such as strength, betweenness centrality, or curvature, specific node(s) and all related edges can be chosen for deletion (i.e., by removing the corresponding row(s) and the column(s)) from the connectivity matrix. Independent graph measures such as connectedness, global efficiency, or entropy can then be computed on the new ("altered") connectivity matrix. This process is typically performed using decreasing nodal measures (e.g. strength), so that important nodes are deleted first. Those measures are recomputed after each iteration, the nodes are re-ordered accordingly, and the whole process is repeated until all nodes have been deleted.

Traditionally, integration and centrality graph metrics like global efficiency, betweenness centrality, degree centrality, characteristic path length, and clustering coefficient have been used to study the robustness of brain networks[10]. While these measures certainly provide very useful information about the networks, we argue, based on the mathematical properties further detailed in sub-section "Ollivier–Ricci curvature and graph robustness" of the Methods section, that graph curvature and entropy may provide complementary metrics for robustness assessment. To explore this, Fig. 5 displays the changes in the size of the largest component, global efficiency, and topological entropy (see Supplementary Note 6), when nodes are deleted based on decreasing strength, betweenness centrality, or curvature. For this experiment, the average of the 33 MGH-USC HCP Consortium DSI connectivity matrices was used (as described in the "Methods" section). These graphs enable to better understand how each nodal robustness measure (e.g. curvature) relate to and impact global graph metrics. They do not necessarily indicate whether a nodal measure is "better" than another, but rather characterize their (dis)similarities with respect to a particular global metric. For the original connectivity matrix (Fig. 5, top row), we observe that curvature and betweenness centrality show similar behaviors for global efficiency and entropy, while strength leads to a faster decay of these two graph metrics. We could therefore conclude that strength is a "better" measure of robustness. However, looking at the size of the largest graph component, strength, centrality, and curvature all behave differently—with centrality leading to a faster decay of the largest component size—thereby illustrating that curvature provides information that is complementary to the other nodal measures.

We note that Alstott et al.[11] used a Gaussian transformation of the connectivity weights for this type of robustness analysis (see discussion below). We therefore present results using the same transformation in Fig. 5 (bottom row). Additionally, Supplementary Note 7 reproduces the results presented in Fig. 3 from Alstott et al.[11], to support the correctness of our analysis. With this

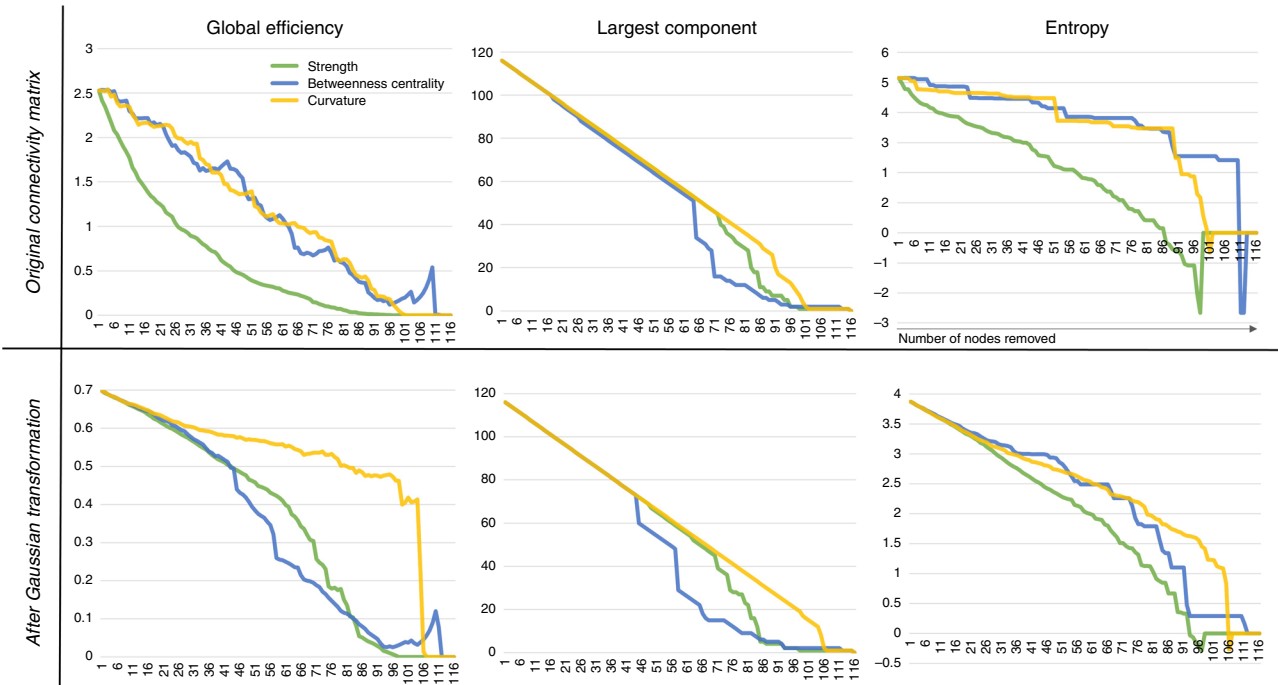

**Fig. 5** Robustness analysis using node deletion for the lower resolution connectivity matrices (116 × 116) generated using the AAL atlas and MGH-USC DSI datasets. The size of the largest component, global efficiency and entropy are computed (with or without transformation of the connectivity matrix weights) after targeted removal of nodes with decreasing strength, betweenness centrality, or curvature. The top row shows results for the original connectivity matrix while the bottom row shows results after Gaussian transformation of its weights as done in Alstott et al.[11]

approach, results stay consistent to those obtained without Gaussian transformation, except for global efficiency. In this case, all three nodal measures behave differently with centrality leading to the fastest decay in global efficiency. We also note that, for each step of such a robustness analysis, as presented in Fig. 5, a new node is deleted and the size of the network is reduced. Thus, we end up comparing parameters across brain networks with different parcellations (i.e. different number of nodes). It is important to remember that network topological measures show strong dependence on the nodal scale[7], therefore, differences between nodal measures may also be attributed, to some extent, to different nodal scales.

In order to further assess the role and unique contribution of curvature, in comparison with other measures, we also computed the Pearson correlation between node measures using all 33 DSI datasets from the MGH-USC HCP Consortium[3,22] (116 nodes using AAL atlas). Correlations were obtained, for each dataset, across the 116 nodal values and their empirical distributions were obtained by repeating this across the 33 datasets (see Supplementary Note 8, where the mean and variance of the histograms are shown on top). Curvature does not present a different behavior than the other node measures. Most importantly, the nodal measures can be seen as only weakly to moderately correlated with each other, with curvature positively correlated to strength and (to a much lesser extent) betweenness centrality, while strength and betweenness centrality also show positive correlation. This result further supports our claim that curvature provides information about brain networks, which is complementary to other graph measures.

**Note on Gaussian transformation of connectivity matrices**. Streamlines numbers (i.e., the weights of structural connectivity graphs) produced by tractography algorithms are exponentially distributed[7,20]. Without altering the rank-ordering of pathways, those distributions can be transformed to follow Gaussian

distributions[11], for ease of analysis (e.g. see entropy decay in Fig. 5, bottom row). However, we would like to emphasize that such a transformation may lead to changes in edges weights (and consequently node strengths) that may affect graph measures differently (see results for global efficiency in Fig. 5), and thereby possibly biasing the definition of important nodes, as well as the identification of graph measures that are adequate to quantify robustness. For instance, the weights, while preserving order, can be mapped to a completely different range of values, thereby increasing or diminishing the relative importance of nodes (e.g. mapping weights in the range [1,1000] to [10,11]). We therefore recommend to apply such transformation(s) with care.

## Discussion
We have introduced the concept of graph curvature to quantify the importance of nodes (meaning that their disruption leads to large changes in the overall graph) in brain networks. We have shown that curvature can indeed help identify important areas (nodes) and points to changes in the brain network structure that may be attributed to age or diseases. The close relation between curvature and network robustness points to the significance of the detected nodes in supporting robust brain functions. Thus, this study lays the foundation for a new approach to assess brain network robustness at the nodal level. We argue that the information provided by curvature may be used in combination with other nodal measures for studying global changes in the brain.

Curvature (averaged across the network) can also provide a global graph measure for the quantification of brain network robustness. A similar viewpoint has recently been proposed in the context of financial networks and of gene-regulatory networks[16,17,54]. It is indeed quite interesting that the connection between curvature and the ability of a dynamical process on a network to return to equilibrium after a perturbation (robustness) is observed in such disparate problems (economy, thermo-dynamics gene regulation, cancer, and now: brain networks).

Several other directions may be worthy of investigation along the same lines. In particular, studying curvature changes between nodes at the edge level may prove particularly effective as, in that case, critical changes in interactions between areas in the brain may be easier to detect. We propose these future directions with the caveat that curvature is sensitive to the way connectivity matrices are generated, i.e., curvature is affected by the choice of parcellation scale, tractography algorithms, as well as the particular type of diffusion data, e.g., DTI, HARDI, DSI, etc. Therefore, care must be exercised to minimize such possible effects. The present work focused mainly on exploring the concept of node curvature as a measure of robustness of brain structural networks, in comparison with existing measures.

## Methods

**Overview.** First, we describe generic notions of distance and curvature on metric spaces (i.e., Riemannian manifolds). These concepts are needed to understand how brain networks (e.g., graphs) curvature and robustness can be characterized. Next, we describe how curvature can be defined and computed on discrete spaces, such as brain networks with finite (and usually low) number of nodes. Finally, we relate curvature to robustness, and explain how it can be efficiently computed and used to assess the ability of a graph to withstand perturbations.

**Wasserstein distance and optimal mass transport.** Let $p$ and $q$ be two probability distributions on the discrete metric space $\mathscr{X}$ equipped with metric $d(\cdot, \cdot)$. The transportation cost of a unit mass from point $x_i \in \mathscr{X}$ to $x_j \in \mathscr{X}$ is denoted as $c_{i,j} \geq 0$. Denote by $\pi_{i,j} \geq 0$ the transference plan, i.e., the (probability) measure of the amount of mass transferred from $x_i$ to $x_j$.

The optimal mass transportation (OMT) problem seeks an optimal transference plan ($\pi$) that minimizes the total cost of moving $p$ to $q$. This can be formulated as the following optimization problem[55–57]:

$$\min_{\pi} \sum_{i,j} c_{i,j} \pi_{i,j} \qquad (1)$$

$$\text{subject to} \sum_{j} \pi_{i,j} = p_i, \quad \forall i$$

$$\sum_{i} \pi_{i,j} = q_j, \quad \forall j$$

$$\pi_{i,j} \geq 0 \quad \forall i,j.$$

The problem in Eq. (1) may be expressed in the matrix form:

$$\min_{\Pi \in \mathscr{M}_{(p,q)}} \text{trace}(C^{\mathrm{T}}\Pi) \qquad (2)$$

with

$$\mathscr{M}_{(p,q)} := \{\Pi | \Pi \mathbb{1} = p, \Pi^T \mathbb{1} = q, \Pi \geq 0\},$$

$$C = [c_{i,j}], \Pi = [\pi_{i,j}].$$

Here $\mathbb{1}$ is the column matrix of ones with the appropriate dimension.

When the cost $c$ is defined as $c_{i,j} = d(x_i, y_j)^r$, for any positive integer $r$, we can define the $r$-Wasserstein distance[56,58] as

$$W_r(p,q) := \left( \min_{\Pi \in \mathscr{M}_{(p,q)}} \text{trace}(C^{\mathrm{T}}\Pi) \right)^{1/r}. \qquad (3)$$

When $r = 1$ this is also known as the earth mover's distance (EMD). We will use this version of OMT in the present work.

**Generalities on curvature.** In this section, we introduce the key notion of curvature from Riemannian geometry. For $X$ an $n$-dimensional Riemannian manifold, $x \in X$, let $T_x$ denote the tangent space at $x$, and $v_1, v_2 \in T_x$ orthonormal tangent vectors. Then, for geodesics (local curves of shortest length) $\alpha_i(t) := \exp(tv_i)$, $i = 1, 2$, the sectional curvature $K(v_1, v_2)$ measures the deviation of geodesics relative to Euclidean geometry, i.e.,

$$d(\alpha_1(t), \alpha_2(t)) = \sqrt{2}t\left(1 - \frac{K(v_1, v_2)}{12}t^2 + O(t^4)\right). \qquad (4)$$

The Ricci curvature is the average sectional curvature. In other words, given a (unit) vector $v \in T_x$, we complete an orthonormal basis $v_1, v_2, \ldots, v_n$, and define $Ric(v) := \frac{1}{n-1}\sum_{i=2}^{n} K(v, v_i)$. The Ricci curvature may be extended to a quadratic form known as the Ricci curvature tensor[59]. The scalar curvature is then defined to be the trace of this quadratic form.

There are a number of alternative characterizations of Ricci curvature[59]. In this paper, we employ the following definition: referring to Fig. 6, consider two very close points $x$ and $y$ in $X$ and associated tangent vectors $w$ and $w'$, where $w'$ is obtained by parallel transport of $w$ along a geodesic (in the direction $v$) connecting the two points. Now, geodesics drawn from $x, y$ along $w, w'$ will get closer when the curvature is positive (positively curved space). This is also reflected in the fact that the distance between two small (geodesic balls) is less than the distance of their centers. The Ricci curvature $Ric(v)$ along the direction $v$ connecting $x, y$ quantifies this contraction. Similar considerations apply to negative and zero curvature.

**Curvature and entropy.** In this section, following the previous studies[16,17], we establish the relationship between curvature and robustness. We consider $X$ to be a complete, smooth Riemannian manifold. The relations may then be extrapolated to discrete spaces.

We denote the space of probability densities by $\mathscr{P}(X)$. Then, one defines the Boltzmann entropy of $\rho \in \mathscr{P}(\mathscr{X})$ as

$$S(\rho) := -\int_X \rho \log \rho \, dm, \qquad (5)$$

where $dm$ denotes the volume measure on $X$. (There are several technical assumptions that should be made to ensure the existence of $S$, see refs. [19,60].)

We can then express the following remarkable result from Lott and Villani[19] and Sturm[60]: Let Ric denote the Ricci curvature defined on $X$, and suppose the $Ric \geq k$ for any tangent vector on $X$. Then, for every $\rho_0, \rho_1 \in \mathscr{P}(X)$, the constant speed geodesic $\rho_t$ with respect to the Wasserstein 2-metric connecting $\rho_0$ and $\rho_1$ satisfies

$$S(\rho_t) \geq (1-t)S(\rho_0) + tS(\rho_1) + \frac{kt(1-t)}{2}W_2(\rho_0, \rho_1)^2, \quad 0 \leq t \leq 1. \qquad (6)$$

In fact, relation (6) implies that $Ric \geq k$[61] (see Theorem 1.1).

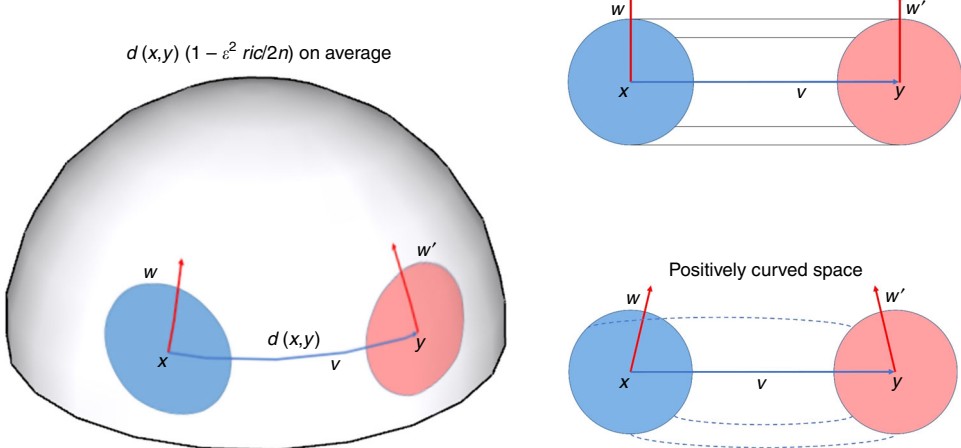

**Fig. 6** In a space with positive Ricci curvature, parallel geodesics emanating from points $x$ and $y$, e.g., in directions along tangent vectors $w$ (at $x$) and $w'$ (at $y$), are drawn closer. In a Euclidean space, the distance between points moving along parallel geodesics at constant speed remains constant

In the works[19,60], Eq. (6) is used as a definition for defining curvature on more general metric measure spaces. It will also motivate an interesting analog of curvature based on the following result[61] (see Theorem 1.5, property (xii)). We now take $X$ to be compact of dimension $n$, and let $A$ denote any measurable subset of $X$. Let $B_r(x)$ denoted the ball of radius $r$, centered around $x$. Then one can define the normalized Riemannian uniform distribution as follows:

$$m_{r,x}(A) = \frac{m(A \cap B_r(x))}{m(B_r(x))}.$$

Then[61], we have that $Ric \geq k$ if and only if

$$W_1(m_{r,x}, m_{r,y}) \leq \left(1 - \frac{k}{2(n+2)}r^2 + o(r^2)\right)d(x,y). \quad (7)$$

Here $W_1$ denotes the Wasserstein 1-metric or EMD. As we will see, Eq. (7) is the basis of the definition due to Ollivier[62–64] of curvature on a weighted graph. But the basic idea is that on a compact, smooth Riemannian manifold conditions (6) and (7) are equivalent, and Eq. (6) indicates the correlation of changes in entropy (defined along geodesic paths) and Ricci curvature. We will express this as

$$\Delta S \times \Delta Ric \geq 0. \quad (8)$$

This is of course restricted to the smooth (Riemannian) case. But the point is that the results just quoted from ref. [61] indicate a deep relationship between changes in entropy and the Ricci curvature as characterized by Eq. (7).

**Ollivier–Ricci curvature**. Ollivier–Ricci curvature or coarse Ricci curvature is the discrete analog of the Ricci curvature[62–64]. Let $(X, d)$ be a geodesic metric space equipped with a family of probability measures $\{p_x : x \in X\}$. We define the Ollivier–Ricci curvature $\kappa(x, y)$ along the geodesic connecting $x$ and $y$ as

$$W_1(p_x, p_y) = (1 - \kappa(x,y))d(x,y), \quad (9)$$

where $W_1$ denotes the earth mover's distance (Wasserstein 1-metric), and $d$ the geodesic distance on the space. Note the similarity to Eq. (7).

For the case of an undirected weighted graph (e.g. a brain structural connectivity network) $G = (V, E, W)$, where $V$ is the set of vertices (nodes), $E$ the set of edges, and $W$ the set of edge weights, we let

$$d_x = \sum_{y \in \mathcal{N}(x)} w_{xy}$$

$$p_x(y) := \frac{w_{xy}}{d_x},$$

where $\mathcal{N}(x)$ denotes the set of nodes that are adjacent to $x$; throughout, we assume that all the edge weights $w_{xy} = w_{yx} \geq 0$ and that $w_{xy} = 0$ if $d(x,y) \geq 2$, or equivalently, if $y \notin \mathcal{N}(x)$. Note here that the geodesic distance $d(x,y)$ is taken to be the hop distance between node $x$ and $y$, i.e., the minimum number of steps it takes to go from $x$ to $y$.

**Node curvature**. The (scalar) node curvature for node $x$ ($\kappa_x$) in the graph is computed by summing the curvature between node $x$ and all its neighboring nodes, i.e.,

$$\kappa_x := \sum_{y \in \mathcal{N}(x)} \kappa(x, y). \quad (10)$$

We also note that an alternative "weighted" version of the node curvature may be defined as

$$\tilde{\kappa}_x := \sum_y p_x(y)\kappa(x,y). \quad (11)$$

**Robustness and the Fluctuation Theorem**. We now turn to the notion of robustness, which we employ in this paper, and subsequently make the link between robustness and curvature. It is based on ideas from statistical mechanics and, in particular, the Fluctuation Theorem formulated in Demetrius et al.[18]. The Fluctuation Theorem measures the ability of a network to maintain its functionality in the face of perturbations (internal or external).

Let $p_\delta(t)$ be the probability that the mean value of an observation (for a given network) deviates from its original value, by more that $\delta$ at time $t$, due to some perturbation. The rate $R$ at which the system returns back to its original state is defined as

$$R := \lim_{t \to \infty} \left(-\frac{1}{t} \log p_\delta(t)\right). \quad (12)$$

Note that large $R$ means not much deviation and small $R$ implies a large deviation. In statistical mechanics, it is well-known that entropy and rate functions from large deviations are very closely related[18,65]. The Fluctuation Theorem is a mathematical statement relating the positive correlation of changes in system

entropy $\Delta S$ to changes in robustness $\Delta R$:

$$\Delta S \times \Delta R \geq 0. \quad (13)$$

**Ollivier–Ricci curvature and graph robustness**. Based on the equivalence of Eqs. (6) and (7), in this paper we employ Olliver–Ricci curvature as a proxy for network entropy and thus via the Fluctuation Theorem for network robustness as was proposed in refs. [16,17]. This is of course not a theorem, but a useful analogy. We express this "positive correlation" of graph Ricci curvature and robustness as follows:

$$\Delta R \times \Delta Ric \geq 0. \quad (14)$$

Once again, we emphasize that this is an extrapolation, not a theorem, based on the results from continuous geometry. There are a number of other reasons to see that curvature does indeed have a connection to network robustness which we enumerate here:

1. *Invariant triangles*: The Ollivier–Ricci curvature can be viewed as feedback measure, i.e., the number of triangles in a network (redundant pathways) can be characterized by an explicit lower bound based on Ollivier–Ricci curvature; see Theorem 2 of ref. [66]. Feedback redundancy is a key measure of system robustness.
2. *Stochastic systems*: Ollivier[62] studied this notion of curvature for the Ornstein–Uhlenbeck stochastic process; see Example 9. As noted in ref. [16], this gives a direct correlation of changes in the rate function (12) and Olliver–Ricci curvature (see pp. 10–11).
3. *Convergence to equilibrium*: Positive Ollivier–Ricci curvature controls the rate of convergence to the invariant (equilibrium) distribution on a weighted graph (Markov chain) and the larger the curvature the faster the rate; see Corollary 21 of ref. [62]. This is another indication of the connection of curvature to robustness.

Since curvature can easily be computed via linear programming[55,57], it provides an attractive and novel tool to study the robustness of networks represented as weighted graphs, such as brain connectivity networks. In the next section, we briefly summarize existing measures to characterize complex brain networks and provide information about the datasets which we used to demonstrate the benefits of curvature.

**Measures of brain networks characteristics**. We hereafter briefly summarize important graph-theoretical measures, which have been introduced to characterize brain networks[67], and are used in our experiments.

1. *Node strength* ($s_i$): The strength of a node $i$ is the sum of the weights $w_{ij}$ of the node's adjacent edges[68], i.e.,

$$s_i = \sum_{j \in \mathcal{N}(i)} w_{i,j}. \quad (15)$$

dMRI data may be employed, in combination with deterministic or probabilistic propagation methods of vector fields (called tractography), to assess the likelihood of connections between cortical and sub-cortical areas[69]. Such likelihood can be obtained by the number of three-dimensional curves generated by these integration or propagation methods and used, in the context of brain structural networks, to define the weight $w_{i,j}$ of an edge between two nodes $i$ and $j$. This summarizes how strongly connected those nodes are to each others, and to the rest of the brain.

2. *Betweenness centrality* ($g_i$): The betweenness centrality of a given node $i$ is defined as the number of shortest paths between pairs of nodes that pass through the node $i$[70], i.e.,

$$g_i = \sum_{i \neq j \neq k} \frac{\sigma_{j,k}(i)}{\sigma_{j,k}}. \quad (16)$$

where $\sigma_{j,k}$ is the total number of shortest paths from node $j$ to node $k$ and $\sigma_{j,k}(i)$ is the number of those paths that pass through node $i$.

3. *Clustering coefficient* ($C_i$): The clustering coefficient of node $i$ is a measure of the density of connections among the node's topological neighbors[71,72]. This is defined as follows: Take $i \in V$, the vertex set of a graph $G = (V, E, W)$, and assume unit weights $e_{ij} \in W$ for all existing edges. Suppose that node $i$ has $k_i$ neighbors. For an undirected graph (which is usually the case for brain structural connectivity networks), there can be at the most $k_i(k_i - 1)/2$ edges in the local neighborhood. Then, $C_i$ is defined as the fraction of the edges that actually exist between the immediate neighbors of $i$ over the maximal number of edges, i.e.,

$$C_i = \frac{2|\mathcal{N}(i)|}{k_i(k_i - 1)}. \quad (17)$$

As before, $\mathcal{N}(i) = \{j : e_{ij} \in E\}$ is the set of immediate neighbors of $i$, and $|\mathcal{N}|$ denotes the cardinality of this set.

**Family-wise error correction**. For the results shown between age groups and in ASD, related to structural changes, family-wise error rate (correction for multiple

comparisons) was controlled using the Holm–Sidak[26] method with $\alpha = 0.05$. The correction was done in GraphPad Prism 8 (https://www.graphpad.com/scientific-software/prism/), assuming data was sampled from normal distributions with identical standard deviations (homoscedasticity assumption) when computing the two-sided $p$ values. The number of unpaired $t$-tests corrected for was equal to the number of nodes i.e., 333 for Gordon atlas.

**dMRI datasets**. As briefly described in the section "Introduction", we used five different public open access datasets in our experiments, and we now provide more details about this data. First, we analyze the high-resolution connectivity matrices created and analyzed by Hagmann et al.[20], using DSI data from five healthy subjects. These matrices are available from the USC Multimodal Connectivity Database[21,22], which enables the reproduction of the original results[20] and evaluation of our method with the exact same datasets (more specifically the ability of node curvature to capture novel information). We also analyze 33 new DSI datasets obtained from the MGH-USC HCP Consortium[3,22], to demonstrate the consistency of our findings on critical brain areas. Next, our experiments use high angular resolution diffusion imaging (HARDI) datasets obtained from the WU-Minn HCP Consortium Lifespan Pilot Project[1] (https://www.humanconnectome.org/study-hcp-lifespan-pilot) to illustrate the ability of node measures to capture changes in certain brain areas, which are related to age. Finally, our last experiments are performed with DTI datasets from ABIDE-II[37] (http://fcon_1000.projects.nitrc.org/indi/abide/abide_II.html), to investigate differences in brain structural connectivity in ASD.

1. *DSI Datasets from Hagmann et al.*: Data was acquired from five healthy right-handed male subjects, on a Philips Achieva 3T scanner with voxel size $2 \times 2 \times 3\,mm^3$, TR/TE = 4200/89 ms and 129 diffusion gradients with a maximum $b$-value of $9000\,s\,mm^{-2}$, for a total acquisition time of 18 min. After segmentation of the white and gray matter, 998 cortical regions-of-interest were created, with an average size of $1.5\,cm^2$. Tractography was then performed, and structural connectivity matrices created by defining the weight of each edge as the number of streamlines per unit surface (i.e. density). Additional details can be found in the original paper[20].

2. *DSI datasets from the MGH-USC HCP consortium*: Data was acquired from 35 healthy adults (age range 20–59) scanned on the customized Siemens 3T Connectom scanner and are available at https://db.humanconnectome.org. Two of the datasets were not included in our experiments because of pre-processing errors in our analysis pipeline. Acquisition parameters included voxel size of $1.5 \times 1.5 \times 1.5\,mm^3$, TR/TE = 8800/57 ms and four $b$-values (with corresponding number of diffusion gradients in parenthesis): $1000\,s\,mm^{-2}$ (64), $3000\,s\,mm^{-2}$ (64), $5000\,s\,mm^{-2}$ (128), $10,000\,s\,mm^{-2}$ (256), for a total acquisition time of about 89 min. Connectivity matrices were generated using DSI Studio (http://dsi-studio.labsolver.org) as described below.

3. *HARDI Datasets from the WU-Minn HCP Consortium*: Lifespan data was acquired from healthy subjects across the lifespan in six age groups: 4–6, 8–9, 14–15, 25–35, 45–55, and 65–75 years and are available at: https://db.humanconnectome.org. We analyzed the data acquired on the UMinn Siemens 3T Prisma scanner (Phase 1a), which include five participants per age group (ages 25–35, 45–55, and 65–75) or six participants per age group (ages 8–9 and 14–15). Acquisition parameters included voxel size of $1.5 \times 1.5 \times 1.5\,mm^3$, TR/TE = 3222/89 ms and two $b$-values, $1000\,s\,mm^{-2}$ and $2500\,s\,mm^{-2}$, each with 75 diffusion gradients acquired twice with opposite phase-encoding polarity, for a total acquisition time of about 22 min. Connectivity matrices were also generated using DSI Studio (http://dsi-studio.labsolver.org) as described below.

4. *DTI datasets from ABIDE-II*

   a. *Trinity Center for Health Sciences ASD Data*: Data was acquired from 20 typically developing control subjects (15–20 years) and 20 subjects with ASD (10–20 years) using a Philips Intera Achieva 3T system. Acquisition parameters included voxel size of $2 \times 2 \times 2\,mm^3$, TR/TE = 20244/79 ms and $b$-value $1500\,s\,mm^{-2}$ with 61 diffusion gradients, for a total acquisition time of about 24:21 min. Connectivity matrices were also generated using DSI Studio (http://dsi-studio.labsolver.org) as described below. Additional details can be found in the original paper[73].

   b. *San Diego State University ASD data*: Data was acquired from 24 typically developing control subjects (8–18 years) and 29 subjects with ASD (7–18 years) using a GE 3T MR750 system. Acquisition parameters included voxel size of $1.875 \times 1.875 \times 2\,mm^3$, TR/TE = 8500/84.9 ms and $b$-value $1000\,s\,mm^{-2}$ with 61 diffusion gradients. Connectivity matrices were also generated using DSI Studio (http://dsi-studio.labsolver.org) as described below. Details can be found at the ABIDE-II website http://fcon_1000.projects.nitrc.org/indi/abide/scan_params/ABIDEII-SDSU_1_scantable.pdf.

1. *HCP-DSI dataset*: To run tractography and generate connectivity matrices for the DSI data, seeds were placed randomly in the whole brain with the following settings: normalized quantitative anisotropy (NQA) threshold: 0.08, angular threshold: $60°$, tractography method: Runge–Kutta[75], total number of streamlines: 5 million. (Although similar results were obtained with $500,000$ streamlines, we used 5 millions to ensure consistency with previous work[20].) 116 cortical areas (nodes) were automatically segmented via non-linear registration of the automated anatomical labeling (AAL) template available in DSI Studio. Connectivity matrices were constructed with weights defined as the number of streamlines connecting each pair of cortical areas (nodes).

2. *HCP-HARDI dataset*: For the HARDI data, diffusion tensors were estimated to perform deterministic tractography. Seeds were also placed randomly in the whole brain with the following settings: fractional anisotropy (FA) threshold: 0.1, angular threshold: $60°$, tractography method: Runge–Kutta[75], total number of streamlines: $500,000$. 333 cortical areas (nodes) were automatically segmented via non-linear registration of the Gordon cortical template[25] available in DSI Studio. Connectivity matrices were constructed with weights defined as the number of streamlines connecting each pair of cortical areas (nodes). Node numbers (IDs), centroid and community (group) of each node/area parcellation of 333 cortical parcellations from resting-state fMRI can also be downloaded from https://sites.wustl.edu/petersenschlaggarlab/parcels-19cwpgu/[25].

3. *ABIDE-II datasets (SDSU and TC ASD data)*: Settings for the generation of connectivity matrices for both datasets were identical to those used for the HCP-HARDI dataset, including the use of 333 cortical areas (nodes) which were segmented via non-linear registration of the Gordon template[25] available in DSI Studio. Node numbers (IDs), centroid and community (group) of each node from the 333 cortical parcellations from resting-state fMRI can also be downloaded from https://sites.wustl.edu/petersenschlaggarlab/parcels-19cwpgu/[25].

**Reporting summary**. Further information on research design is available in the Nature Research Reporting Summary linked to this article.

## Data availability
The diffusion MRI datasets used in this study are publicly available in the following repositories: DSI Datsets from Hagmann et al [https://doi.org/10.1371/journal.pbio.0060159][20], USC Multimodal Connectivity Database (http://umcd.humanconnectomeproject.org/), HARDI Datasets WU-Minn HCP Consortium Lifespan (https://www.humanconnectome.org/study-hcp-lifespan-pilot[1]) and ABIDE-II ASD Datasets (http://fcon_1000.projects.nitrc.org/indi/abide/abide_II.html[37]).

## Code availability
The code written in MATLAB is available upon request.

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

## Acknowledgements

This work was partly supported by AFOSR grant FA9550-17-1-0435 (T.T.G., A.T.), NIH grants P41 EB015894 (C.L.), P41 EB027061 (C.L.), P30 NS076408 (C.L.), P41 EB015902 (A.T.), U24 CA180924 (A.T.), R01 AG048769 (A.T.), NSF grant 1665031 (H.F., Y.C., T.

T.G.) and the Fulbright Program (H.F.). Data were provided in part by the Human Connectome Project, WU-Minn Consortium (Principal Investigators: David VanEssen and Kamil Ugurbil; 1U54MH091657) funded by the 16 NIH Institutes and Centers that support the NIH Blueprint for Neuroscience Research; and by the McDonnell Center for Systems Neuroscience at Washington University. Data were also provided in part by the Human Connectome Project, MGH-USC Consortium (Principal Investigators: Bruce R. Rosen, Arthur W. Toga, and Van Wedeen; U01MH093765) funded by the NIH Blueprint Initiative for Neuroscience Research grant; the National Institutes of Health grant P41EB015896; and the Instrumentation Grants S10RR023043, 1S10RR023401, 1S10RR019307. We thank Dr. Patric Hagmann for making the connectivity matrices, published in ref. [20], freely available via the USC MultimodalConnectivity Database. We also would like to thank Dr. Eric F. Lock from the Division of Biostatistics, School of Public Health, University of Minnesota, for his input with some of the statistical analyses.

## Author contributions

H.F., T.T.G., A.T. and C.L. have equal contribution in conceiving the method. Y.C. and H.F. wrote the manuscript and implemented the algorithm in MATLAB. H.F. wrote the Supplementary Information. All authors have read and approved the final manuscript.

## Competing interests

The authors declare no competing interests.
