## [Peer Review File · Nature Communications]

Reviewers' comments:

Reviewer #1 (Remarks to the Author):

"Network curvature," by Farooq et al advances a novel measure of brain network topology that is of relevance to network centrality and robustness to attack. The strengths of the paper include application to various new and legacy imaging data sets, and exploratory applications to aging and autism data. The new method might be of interest to those using graph theory to understand structural and functional brain networks although its broader appeal is not readily apparent. Several substantial technical limitations require attention before the paper could be considered for publication:

1. According to the last paragraph of the Results, all contrasts were conducted (over large numbers of network nodes) "for these exploratory results" with an uncorrected p-value of 0.05. Given the number tests performed, many (an unknown number) of the reported results will be false positives.
2. The contrasts between autism and ageing yield nodes that are interpreted as showing that curvature yields a more sensitive measure of group differences. However, without prior hypotheses, a better prior appraisal of the literature and proper control for family-wise error these new "hits" could equally be false positives and provide few novel insights.
3. The first paragraph of the Intro is a didactic introduction to the field of connectomics and does little to focus on specific problems that require to be addressed or the context of the present problem.
4. A considerable number of the results, including all the robustness analyses, are presented in the Discussion.
5. While the mathematical description of the new metric is very nicely presented, insufficient attempt is given to explain the intuitive meaning of the metric. This, taken together with the preceding problems, gives the paper an opportunistic feel – that is the application of a nice measure taken from another field into neuroscience but without purpose or context, rather simply because it has been applied elsewhere.

My recommendation is for the authors to repeat the paper with proper family-wise error control, consider edits and restructuring to better motivate and position the metric, and draw from the relevant literature to more informatively position their clinical contrasts in line with prior knowledge and outstanding issues in those fields.

Reviewer #2 (Remarks to the Author):

Title: Network Curvature as a Hallmark for Brain Structural Connectivity
Authors: Hamza Farooq, Yongxin Chen, Tryphon T. Georgiou, Allen Tannenbaum and Christophe Lenglet

Verdict: Major revision

Summary

In this paper the authors employ a notion of discrete curvature to analyze the robustness of brain networks. The discrete curvature used is the recently developed Ollivier-Ricci curvature which is related to the optimal transportation distance between probability measures on a geometric space. In this paper, as is done in most papers using this notion of curvature for graphs, the distance between nodes is given by the shortest path length and the probability measures correspond to the random-walk measures on the graph.

The authors compute the curvature of nodes in both high and low-resolution brain networks and compare these to two node measures used to classify robustness, strength and betweenness centrality. They use figures to show that nodes with high curvature are often different from nodes with high strength or betweenness centrality. In particular they show that curvature can identify nodes related to lesions.

Next, curvature, strength, betweenness centrality and clustering, are computed for nodes in brain networks of two healthy age groups (8-9) and (65-75). The authors compare the changes of these nodes measures between both age groups and find that nodes with statistically significant change in curvature are different from those nodes identified by the other measures. In particular, curvature identifies changes in areas that have been known to be related to age-related structural changes.

Finally, the authors compare the above mentioned node measures between the brain networks of typically develop (TD) individuals and individuals with Autism Spectrum Disorders (ASD). The results show that curvature identifies differences between nodes that are not identified by the other three measures, and hence that curvature can be helpful in the future to provide new insights into difference between brain connectivity of TD and ASD individuals.

Review

The concept of graph curvature has been receiving a great deal of attention since the seminal paper by Yann Ollivier and has been applied to several areas in network science. As far as I could find, the notion of graph curvature has not yet been used on brain networks and hence can potential help us to better understand their structure and function. In particular I really like the idea to use curvature to detect and identify structural changes in TD and ASD brain networks as well as to understand the structural changes our brains undergo as we age. Overall, the paper is also well-written and has a clear structure. I also want to emphasize that the authors use different data sets for each of the different analysis they perform which I find to be an outstanding practice.

However, apart from the great and novel ideas regarding the implementation of curvature in brain networks and the different datasets used, the manuscript suffers from several issues, both in its theoretical framework and in the execution of the analysis.

Theory

The authors use a relation between curvature and robustness as their main theoretical motivation to consider Ollivier-Ricci curvature in brain networks. In particular they use equation (11) and results from two papers to conclude that there is a positive correlation between the change in curvature and robustness. There are two main problems with this line of reasoning:

1. On page 8 the authors state equation (11) as a result, referring to:

[25] *Lott, J. & Villani, C. Ricci curvature for metric-measure spaces via optimal transport*

[52] *Sturm, K. T. On the geometry of metric measure spaces*

The problem is that the authors never explicitly mention the theorems in these papers which forces the reader to either believe them or go and read both papers. I was unable to find a result in [52] which resembles the claim made in this paper. In [25] I did find a related expression. However, this was a definition [Definition 0.7 equation (0.9)] and not a result. The paper did include a theorem [Theorem 0.12] relating this definition to the Ricci curvature but it was completely unclear to me how the definition for the Ricci curvature from [25] is related to that of Ollivier, which is used in this manuscript.

2. The next statement on page 8 uses results from

[22] *Sandhu R. S. et al. Graph curvature for differentiating cancer networks*

[23] *Sandhu R. S., Georgiou T.T. & Tannenbaum, A. R. Ricci curvature: An economic indicator for market fragility and systemic risk*

to establish the positive correlation between the change in curvature and robustness, equation (13). I investigated the reasoning in [22], found on page 10, where this relation is derived [equation (18) there]. The problem here is that they first define equation (11) [equation (17) there] and then simply state that this definition implies the positive correlation (13). Clearly, this reasoning is not sound, or at the very least not well motivated.

Finally, I want to stress that the implementation of Ollivier-Ricci curvature on graphs, as presented in this manuscript and many other works, is just one of many possible implementations, since many different probability measures on graphs or even metrics can be considered. Moreover, it has, in general, no relation to notions of curvature on manifolds. In particular, although Proposition 6 in *Ollivier, Y. Ricci curvature on Markov chains on metric spaces* and the following Example 7 establish a relationship between Ollivier-Ricci curvature and the Ricci curvature on general smooth Riemannian manifolds, this relation is asymptotic as both the distance between the two points as well as the random walk radius go to zero. This can no longer be true in the metric space of graphs with shortest path length, since here the distance is always greater or equal to 1. Furthermore, even when we consider a random geometric graph on a Riemannian manifold where the connection radius decreases with the size of the network, it is not clear at all whether such a asymptotic relation between Ollivier-Ricci curvature and the Ricci curvature holds. Therefore I would be very careful in directly relating the curvature considered in this paper to that of manifolds.

Given these three issues I feel that the theoretical foundation driving the motivation of the authors to consider Ollivier-Ricci curvature for robustness of brain networks are strongly overstated. I appreciate the strength of high-level intuition to drive science forward and completely understand the reason why the authors think that Ollivier-Ricci curvature might be an interesting measure for structural connectivity of brain networks. However, I would strongly advice the authors to be clear and truthful in explaining this reason and intuition, emphasizing the places where it has a rigorous base (referring to theorems) and those where results and relations in related fields are extrapolated. This not only strengthens the manuscript as a whole, but also helps other researchers who might use the results presented here to further investigate curvature in the brain and other networks.

Analysis

As far as I understand, the analysis done in this manuscript has three purposes:

1. Show that curvature is a different measure than strength and betweenness centrality
2. Show that curvature can identify differences in connectivity between different age groups that are relevant with respect to robustness of the brain
3. Show that curvature can identify nodes that are different for TD and ASD individuals.

I will briefly comment on each of these:

1. The authors use two different types of data to investigate whether curvature can identify different nodes than the other measures using high resolution data (5 participants) and low resolution data (33 participants). The problem is that their analysis compares each fixed node over the different participants. Hence, the high resolution data is not large enough to make any statistical conclusion. I would therefore advice them to start with the low resolution data and then include a single small paragraph where they mention that their analysis on the high resolution data, although unfit for statistical analysis for fixed nodes on a participant-to-participant basis, shows similar results. On the other hand, if the authors really want to show that curvature is statistically different from other measures then they could also compute the correlation between these measures based on the data for each node in a single participant. This gives a single correlation value per participant and hence, using all 33 participants in the low resolution data, an empirical distribution of this correlation. If this distribution is concentrated around zero then this would be a very strong indication that curvature of a node is completely different from the other measures. This would be a great contribution to the field since it is well known that most nodal measures in the brain are correlated.
2. For the second analysis the authors investigate if curvature can highlight changes in the brain as it ages. However, for this they perform a cohort study (comparing different age groups) and not a longitudinal study, where the brains of the same group of participants are measured over a certain time span. Therefore I believe the claim that the analysis is concerned with changes in the brain due to healthy aging is too strong and should be changed to reflect that the authors compare different age groups of independent participants.

I also want to point out that the figures provided are extremely difficult to read. This is due to the fact that each view-point of the brain displays all nodes, making some of

them overlap so that their labels can't be read, as well as the fact that some nodes are embedded in the brain figure, which also makes the labels hard to read. This problem is especially present in a printed version, but even on a pdf one needs to zoom to 200 percent to get a decent, but still lacking image. I would therefore strongly encourage the authors to redesign their images so that they help the analysis performed in this manuscript instead of creating confusion.

A different problem arises with the second figure (Figure 5) which is said to show nodes with significant difference identified by multiple measures. It is completely unclear from both the text and the figure what this means. In particular, if a node is identified by both curvature and strength, then why are some nodes in Figure 5 not visible in Figure 4? I would advice the authors to be more specific about the differences between these two figures.

3. The ASD analysis suffers from the same two issues with the figures as described above.

Summary

The application of Ollivier-Ricci curvature for analyzing structural connectivity in brain networks is novel and potentially very promising. However, the current theoretical foundation and reasoning behind the use of this curvature is not well motivated and lacks scientific rigor in some places. The analysis uses different data sets for each of the different analyses performed, which is a good practice. However, the analysis used to establish that curvature is different from strength and betweenness is not very convincing, especially for the high resolution data set of only 5 participants. In addition, the figures used are difficult to read and in some cases it is unclear what is actually displayed. Since these figures are heavily used to support the conclusions they should be improved to help the reader understand these conclusions.

I therefore do not recommend this paper for publication at this moment, but strongly encourage the authors to revise it, taking my comments into account, and submit this new version. Since I think all the basic ingredients are here I would like to briefly summarize the three aspects of the paper and how I would like to see these in the new version.

- Argue in a completely transparent and truthful manner, using results for the relation between robustness and curvature on manifolds, that from a theoretical perspective Ollivier-Ricci curvature might be an interesting measure to consider in brain networks to analyze its robustness.
- Use per-node correlation analysis between curvature and other measures to show that there is a very low (maybe zero) correlation between these measures and conclude that in brain networks node curvature definitely measures something that can not be obtained using current measures.
- Use the analysis of brain networks between different age groups and TD vs ASD to show that curvature can identify nodes which are known, from the literature, to be involved in the robustness and functionality of the brain. Use clear description and figures to support these conclusions.

Response To Reviewers

Network Curvature as a Hallmark of Brain Structural Connectivity

Reviewer 1

Reviewer's comments: According to the last paragraph of the Results, all contrasts were conducted (over large numbers of network nodes) “for these exploratory results” with an uncorrected p-value of 0.05. Given the number tests performed, many (an unknown number) of the reported results will be false positives.

Response: To address the issue, we conducted new experiments with two DTI datasets for Autism Spectrum Disorder (ASD) available at Autism Brain Imaging Data Exchange II (ABIDE-II)¹ on high resolution Gordon atlas². The revised version only presents family-wise error corrected results using Holm-Sidak method.

Reviewer's comments: The contrasts between autism and ageing yield nodes that are interpreted as showing that curvature yields a more sensitive measure of group differences. However, without prior hypotheses, a better prior appraisal of the literature and proper control for family-wise error these new “hits” could equally be false positives and provide few novel insights.

Response: Further literature review was carried out and new family-wise error corrected results have been related to prior studies in the revised version.

Reviewer's comments: The first paragraph of the Intro is a didactic introduction to the field of connectomics and does little to focus on specific problems that require to be addressed or the context of the present problem.

Response: We have addressed this and shortened/focused the first paragraph.

Reviewer's comments: A considerable number of the results, including all the robustness analyses, are presented in the Discussion.

Response: We thank for the comments. We used different datasets for analysis performed in the study to establish consistency of the reported results over a variety of scanning protocols and multiple resolutions of generated graphs.

Reviewer's comments: While the mathematical description of the new metric is very nicely presented, insufficient attempt is given to explain the intuitive meaning of the metric. This, taken together with the preceding problems, gives the paper an opportunistic feel – that is the application of a nice measure taken from another field into neuroscience but without purpose or context, rather simply because it has been applied elsewhere.

Response: Thank you for the positive comment and for the critic to bring out the perspective that the new mathematics afford us. We highlight with the help of theoretical explanation and with experiments/ examples using five different DTI and DSI datasets that Graph Curvature provides a novel insight to brain structural connectivity. We related our findings with previous studies on structural analysis of brain networks robustness³. The key new insight that we underscored is that Graph Curvature captures something beyond local connectivity, and indeed, it captures the ease of linking sites and quantifies the significance of certain nodes in this process.

Reviewer 2

Reviewer's comments: The authors use a relation between curvature and robustness as their main theoretical motivation to consider Ollivier-Ricci curvature in brain networks. In particular they use equation (11) and results from two papers to conclude that there is a positive correlation between the change in curvature and robustness. There are two main problems with this line of reasoning: On page 8 the authors state equation (11) as a result, referring to:

[25] Lott, J. & Villani, C. Ricci curvature for metric-measure spaces via optimal transport

[52] Sturm, K. T. On the geometry of metric measure spaces.

The problem is that the authors never explicitly mention the theorems in these papers which forces the reader to either believe them or go and read both papers. I was unable to find a result in [52] which resembles the claim made in this paper. In [25] I did find a related expression. However, this was a definition [Definition 0.7 equation (0.9)] and not a result. The paper did include a theorem [Theorem 0.12] relating this definition to the Ricci curvature but it was completely unclear to me how the definition for the Ricci curvature from [25] is related to that of Ollivier, which is used in this manuscript.

Response: We have tried to make this clearer in the revision. However, since this part is so crucial (and answers even some of the issues described below), we will answer these concerns in more detail here, repeating the notation of the paper, as well as some of the references. Let X denote a smooth complete Riemannian manifold. One can endow the space of probability densities on X (taken with respect to the volume measure dm) with a natural Riemannian structure⁴. We denote the space of probability densities by $\mathcal{P}(X)$. Then, one defines the **Boltzmann entropy** of $\rho \in \mathcal{P}(X)$ as

$$S(\rho) := - \int_X \rho \log \rho \, dm, \quad (1)$$

where dm denotes the volume measure on X .

Let Ric denote the Ricci curvature defined on X , and suppose the $Ric \geq k$ on X . Then, for every $\rho_0, \rho_1 \in \mathcal{P}(X)$, the constant speed geodesic ρ_t with respect to the Wasserstein 2-metric connecting ρ_0 and ρ_1 satisfies

$$S(\rho_t) \geq (1-t)S(\rho_0) + tS(\rho_1) + \frac{kt(1-t)}{2} W(\rho_0, \rho_1)^2, \quad 0 \leq t \leq 1. \quad (2)$$

(In the mathematics literature quoted below this is stated as the displacement k -convexity of $-S$.) **This relation is the motivating equation for our argument relating curvature and entropy.** In fact, the displacement k -concavity of the entropy given in (2) turns out to be equivalent to lower bounding the Ricci curvature by k .

Now answering your question about the references in Sturm⁵ and Lott-Villani⁶, (2) is used as a **definition** of curvature on more general metric measure spaces and stated as a theorem on a Riemannian manifold. (For example, in Sturm⁵, it is discussed above the inequality on page 69 relating the Hessian and Ricci curvature.) But there is a reference, in which this statement is explicit. Indeed, we have added the reference by von Renesse and Sturm⁷ that makes this explicit. There is also the online version: <https://pdfs.semanticscholar.org/8106/841de1aebf030080a489a8a6f28e4e974487.pdf>. Please see Theorem 1 of the online version. (In the journal version⁷, this is Theorem 1.1.) Referring again to the latter work, the Ollivier-Ricci curvature is a discrete analogue of the property (xii) (equation 2) of Theorem 3 on the same online version, and Theorem 1.5, property (xii) in⁷. We will very briefly state this result here. There is a much more detailed summary in the revised paper.

We now take X to be compact of dimension n , and let A denote any measurable subset of X . Given a ball $B_r(x)$ centered at x of radius r , one can define the *normalized Riemannian uniform distribution* $m_{r,x}$ (definition given in⁷). Then⁷, we have that $Ric \geq k$ if and only if

$$W_1(m_{r,x}, m_{r,y}) \leq \left(1 - \frac{k}{2(n+2)} r^2 + o(r^2) \right) d(x,y). \quad (3)$$

Here W_1 denotes the Wasserstein 1-metric. (3) underpins the intuition of the definition of Ollivier⁸ of curvature on a weighted graph. But the basic idea is that on a compact, connected Riemannian manifold conditions (2) and (3) are equivalent, and (2) indicates the correlation of changes in entropy and Ricci curvature.

As you correctly point out, in the case of a graph, we cannot go to any limit. However, based on this, we can **extrapolate**, and employ a discrete notion of curvature (due to Ollivier) as a proxy for entropy, and therefore robustness (via the Fluctuation Theorem). We have tried to make all of this clear in the present write-up.

Reviewer's comments: The next statement on page 8 uses results from:

[22] Sandhu R. S. et al. Graph curvature for differentiating cancer networks

[23] Sandhu R. S., Georgiou T.T. & Tannenbaum, A. R. Ricci curvature: An economic indicator for market fragility and systemic risk to establish the positive correlation between the change in curvature and robustness, equation (13). I investigated the reasoning in [22], found on page 10, where this relation is derived (equation (18) there). The problem here is that they first define equation (11) [equation (17) there] and then simply state that this definition implies the positive correlation (13). Clearly, this reasoning is not sound, or at the very least not well motivated.

Response: Yes, we agree that this statement is not a theorem, but as we note above, it is an extrapolation/analogy. This is now more clearly stated in the revised manuscript. On a Riemannian manifold displacement k -concavity of the entropy (defined above) is equivalent to an Ollivier-type characterization of the curvature. We therefore use this as a motivation for relating Ollivier-Ricci to entropy. There are several other ways of motivating the connection of curvature to robustness. First of all, as quoted in [22], it has been argued that feedback loops are essential to the functionality of biological mechanisms and systems that arise from deliberate Darwinian-like principles^{9,10}. The Ollivier-Ricci curvature can be viewed as feedback measure, i.e., the number of triangles in a network (redundant pathways) can be characterized by the lower bound of Ricci curvature; see Theorem 2¹¹. Secondly, as noted by Ollivier, one can also see that relationship of robustness to the Ollivier-Ricci curvature in the following manner¹² dealing with Markov chains. The basic idea is that *larger Ollivier-Ricci curvature indicates greater robustness via rate of convergence to the invariant (equilibrium) distribution*; see Corollary 21. Finally, via an explicit computation (again in¹², Example 9), for an Ornstein-Uhlenbeck process, one can explicitly compute that Ollivier-Ricci curvature is positively correlated to the rate function. (For a detailed derivation see [22]; see pages 10-11.)

Reviewer's comments: Finally, I want to stress that the implementation of Ollivier-Ricci curvature on graphs, as presented in this manuscript and many other works, is just one of many possible implementations, since many different probability measures on graphs or even metrics can be considered. Moreover, it has, in general, no relation to notions of curvature on manifolds. In particular, although Proposition 6 in Ollivier, Y. Ricci curvature on Markov chains on metric spaces and the following Example 7 establish a relationship between Ollivier-Ricci curvature and the Ricci curvature on general smooth Riemannian manifolds, this relation is asymptotic as both the distance between the two points as well as the random walk radius go to zero. This can no longer be true in the metric space of graphs with shortest path length, since here the distance is always greater or equal to 1. Furthermore, even when we consider a random geometric graph on a Riemannian manifold where the connection radius decreases with the size of the network, it is not clear at all whether such a asymptotic relation between Ollivier-Ricci curvature and the Ricci curvature holds. Therefore I would be very careful in directly relating the curvature considered in this paper to that of manifolds.

Response: Agreed! As we have argued, Ollivier-Ricci curvature is simply an analogue. On the other hand, as we have just indicated, Ollivier-Ricci curvature does seem to be connected to various other notions of robustness. This has been explicitly enumerated in the revised manuscript in the section on Ollivier-Ricci Curvature and Graph Robustness.

Reviewer's comments: Given these three issues I feel that the theoretical foundation driving the motivation of the authors to consider Ollivier-Ricci curvature for robustness of brain networks are strongly overstated. I appreciate the strength of high-level intuition to drive science forward and completely understand the reason why the authors think that Ollivier-Ricci curvature might be an interesting measure for structural connectivity of brain networks. However, I would strongly advice the authors to be clear and truthful in explaining this reason and intuition, emphasizing the places where it has a rigorous base (referring to theorems) and those where results and relations in related fields are extrapolated. This not only strengthens the manuscript as a whole, but also helps other researchers who might use the results presented here to further investigate curvature in the brain and other networks.

Response: We thank you for this very constructive advice. We have tried to make a more convincing argument to support our use of Ollivier-Ricci as a proxy for robustness on a weighted graph. At the same time we have been careful to be "truthful." It is indeed delicate on how to bring out geometric intuition by analogy, based on mathematical results that only apply under stringent conditions. We hope that we have stricken a reasonable balance.

Reviewer’s comments: The authors use two different types of data to investigate whether curvature can identify different nodes than the other measures using high resolution data (5 participants) and low resolution data (33 participants). The problem is that their analysis compares each fixed node over the different participants. Hence, the high resolution data is not large enough to make any statistical conclusion. I would therefore advice them to start with the low resolution data and then include a single small paragraph where they mention that their analysis on the high resolution data, although unfit for statistical analysis for fixed nodes on a participant-to-participant basis, shows similar results. On the other hand, if the authors really want to show that curvature is statistically different from other measures then they could also compute the correlation between these measures based on the data for each node in a single participant. This gives a single correlation value per participant and hence, using all 33 participants in the low resolution data, an empirical distribution of this correlation. If this distribution is concentrated around zero then this would be a very strong indication that curvature of a node is completely different from the other measures. This would be a great contribution to the field since it is well known that most nodal measures in the brain are correlated.

Response:

1. We start with discussion on previously published results which are from high resolution data and then build the argument/explanation on low resolution data. We maintain that the order of discussing high resolution results before the low resolution helps putting the argument in proper perspective of already published results.
2. We computed correlation between node measures based on the data for each node in a single participant using different datasets. Figure 1 shows histograms (pdfs) of the correlations between node measures using low resolution DSI datasets from the MGH-USC HCP Consortium^{13,14} (116 nodes using AAL atlas). Mean and variance of the histograms are shown on top. Curvature does not show different behavior than other node measures. Most of the nodal measures can be seen as weakly correlated to each other with Curvature positively correlated to Strength and Betweenness Centrality while Strength and Betweenness Centrality also show positive correlation.

Figure 1. Correlation Between Different Node Measures Using Low Resolution MGH Dataset of 33 Individuals

Reviewer’s comments: For the second analysis the authors investigate if curvature can highlight changes in the brain as it ages. However, for this they perform a cohort study (comparing different age groups) and not a longitudinal study, where the

brains of the same group of participants are measured over a certain time span. Therefore I believe the claim that the analysis is concerned with changes in the brain due to healthy aging is too strong and should be changed to reflect that the authors compare different age groups of independent participants.

Response: We have changed the title and discussion of LifeSpan experiments from “Curvature Changes in Healthy Development and Aging” to “Curvature Changes in Different Age Groups of Independent Participants”.

Reviewer’s comments: I also want to point out that the figures provided are extremely difficult to read. This is due to the fact that each view-point of the brain displays all nodes, making some of them overlap so that their labels can’t be read, as well as the fact that some nodes are embedded in the brain figure which also makes the labels hard to read. This problem is especially present in a printed version, but even on a pdf one needs to zoom to 200 percent to get a decent, but still lacking image. I would therefore strongly encourage the authors to redesign their images so that they help the analysis performed in this manuscript instead of creating confusion.

Response: Thank you for the suggestion. Figures for both the experiments (Figure 4 and 5 in the manuscript) have now been redrawn on what we feel is an easier to understand/ view format.

References

1. Martino, A. D. *et al.* Enhancing studies of the connectome in autism using the autism brain imaging data exchange II. In *Scientific data* (2017).
2. Gordon, E. M. *et al.* Generation and Evaluation of a Cortical Area Parcellation from Resting-State Correlations. *Cereb. Cortex* **26**, 288–303 (2014).
3. Hagmann, P. *et al.* Mapping the structural core of human cerebral cortex. *PLOS Biol.* **6**, 1–15 (2008).
4. Otto, F. The geometry of dissipative evolution equation: the porous medium equation. *Comm. Partial. Differ. Equations* **26** (2001).
5. Sturm, K. T. On the geometry of metric measure spaces, I and II. *Acta Math.* **196**, 65–177 (2006).
6. Lott, J. & Villani, C. Ricci curvature for metric-measure spaces via optimal transport. *Annals Math.* **169**, 903–991 (2009).
7. von Renesse, M.-K. & Sturm, K.-T. Transport inequalities, gradient estimates, entropy and ricci curvature. *Commun. on Pure Appl. Math.* **58**, 923–940 (2005).
8. Ollivier, Y. Ricci curvature of markov chains on metric spaces. *J. Funct. Analysis* **256**, 810–864 (2009).
9. Csete, M. E. & Doyle, J. C. Reverse engineering of biological complexity. *Sci.* **295** **5560**, 1664–9 (2002).
10. Kitano, H. Cancer as a robust system: implications for anticancer therapy. *Nat. Rev. Cancer* **4**, 227–235 (2004).
11. Bauer, F., Jost, J. & Liu, S. Ollivier-ricci curvature and the spectrum of the normalized graph laplace operator. *Math. Res. Lett.* **19**, 1185–1205 (2012).
12. Ollivier, Y. Ricci curvature of markov chains on metric spaces. *J. Funct. Analysis* **256**, 810–864 (2009).
13. Fan, Q. *et al.* Mgh–usc human connectome project datasets with ultra-high b-value diffusion mri. *NeuroImage* **124**, 1108 – 1114 (2016). Sharing the wealth: Brain Imaging Repositories in 2015.
14. Setsompop, K. *et al.* Pushing the limits of in vivo diffusion mri for the human connectome project. *NeuroImage* **80**, 220 – 233 (2013). Mapping the Connectome.

Reviewers' comments:

Reviewer #1 (Remarks to the Author):

"Network curvature as a hallmark of brain structural connectivity": Revised manuscript

This paper has improved, with additional details regarding the mathematical derivation of the core measure, the inclusion of more illustrative data and the use of a false discovery-based method of multiple hypothesis testing. Nonetheless there remains several important limitations:

1. In establishing (yet another) graph metric, showing that group contrasts (age, autism) identify a combination of similar and distinct brain regions does not establish that the new method establishes novel insights into structural connectivity in these applications: In particular, what do these new nodes add to our understanding of the disorder – these interpretations are rather thin and lean on reverse inference when the authors do appeal to the reported functions of the observed differences.

To address this, the authors could undertake a suitable multivariate analysis that links subject-wise structural (curvature) network changes with phenotypic differences – this approach has been well established in the community (e.g. using CCA; [1], [2]) and would allow the authors to make more direct and insightful conclusions.

2. Similarly, in judging the role of curvature versus other centrality measures, it would be preferable to perform hierarchical logistic regression to understand, in direct comparison to other measures (and after accounting for some co-linearity) how much variance between the group can be accounted for by curvature.

3. There are other recent measures of node centrality (e.g. controllability [2], fragility[3]) that speak to the same concept as curvature: These should be incorporated into the ms. In addition, controllability – which directly engages with the stability of the system to linear perturbations – likely overlaps substantially and should be benchmarked against curvature.

4. One sentence reporting that statistical correction was performed with the Holm-Sidak method is insufficient: How many tests were corrected for? What is the threshold p-value? What are the other test statistics? Are the data normally distributed? This requires a brief subsection in the Methods. And a minor point is that all the p-values in the supplementary tables are specified to far too high precision (up to 8 significant positions!).

5. It is really challenging to read and interpret the overlays of the small symbols on the surface images (Figure 4 + 5). I remain perplexed about the paragraphs in the Discussion that present new results and refer to new figures – these are anyway tangential to the main story of the paper, do not show anything about curvature, only about node strength normalization and could easily be moved to the supplementary material.

References:

1. Smith SM, Nichols TE, Vidaurre D, Winkler AM, Behrens TEJ, Glasser MF, Ugurbil K, Barch DM, Van Essen DC, Miller KL (2015): A positive-negative mode of population covariation links brain connectivity, demographics and behavior. *Nature neuroscience* 18:1565–1567.
2. Gu, S., Pasqualetti, F., Cieslak, M., Telesford, Q. K., Alfred, B. Y., Kahn, A. E., ... & Bassett, D. S. (2015). Controllability of structural brain networks. *Nature communications*, 6, 8414.
3. Gollo, L. L., Roberts, J. A., Cropley, V. L., Di Biase, M. A., Pantelis, C., Zalesky, A., & Breakspear, M. (2018). Fragility and volatility of structural hubs in the human connectome. *Nature neuroscience*, 21(8), 1107.

Reviewer #2 (Remarks to the Author):

Title: Network Curvature as a Hallmark for Brain Structural Connectivity
Authors: Hamza Farooq, Yongxin Chen, Tryphon T. Georgiou, Allen Tannenbaum and Christophe Lenglet

Verdict: Minor revision

Review

The authors have made great advances in addressing my comments on the previous version of the manuscript. I am happy with their improvement of the theory part, which is now very clear about the rigorous results and intuiting underlying their application of curvature. In particular, I very much like the enumerated list on page 9 on different motivations for the connection between curvature and robustness.

The new figures for the comparison between TD and ASD brains are also much more clear and readable. The authors have also investigated per-node correlations between different measures as I suggested. These indeed show some correlation between curvature and the other two node-centrality measures, in particular curvature and strength. I was however surprised to find that this figure was not included in the manuscript. Since the authors argue that curvature can identify nodes that other measures cannot, these findings should, at the very least be mentioned in the discussion.

The discussion in paper is more a section about using curvature for assessing network robustness. I did not find much "discussion" there at all. Such a comment is also present in the report of the other referee, although the authors do not seem to have acted upon this. I would suggest to properly rename this section and make it very clear what it is about. Here it is important to keep a eye on the storyline of this section, as in the current state the transition between the different subsections is somewhat sudden.

Summary

I am overall very happy with the changes to the manuscript and think it is very close to a publishable state. However, as I mentioned, there are still some small changes needed to make this last step, in particular regarding the discussion section. I would therefore advise the authors to rewrite and rename this section. I also included some small comments below.

Small comments and suggestions

The line numbers are per page and per section.

- p1, Abstract, line (-2): Point (ii) still mentions "healthy aging" while actually different age groups are compared. The corresponding section title has even been changed to properly reflect this. I would therefore recommend making a similar adjustment here.
- p1, Introduction, line 11: "... and, thereby, that reflects..." should be "... and thereby reflects..."
- p3, Results, line (-2): "...identify areas..." should be "...identifies areas..."

- p4, Results, line 7: "patients perform better to detect visual targets..." should be "performs better in detecting visual targets..."
- p4, Results, line (-5): "...literature, is known..." should be "...literature, are known..."
- p4, Discussion, line (-8): "...betweenness centrality can...node importance than strength..."
It is unclear why curvature was not included in this analysis, especially since the whole goal of the paper was to use it as a measure of node importance. Please either explain its absence or add the data to the figures.
- p5, Conclusion, line 1: "In summary," can be removed.
- p9, Ollivier-Ricci Curvature and Graph Robustness, line 10: I am not quite sure if "in" is the right word in "...studies his notion of curvature in the Ornstein-Uhlenbeck...". Shouldn't it be either on or for?
- p10, Measures for Brain Network Characteristics, line 3: "Suppose that the node i ..." should be "Suppose that node i ..."
- p10, Diffusion MRI Datasets: Here there are three instances where the use of curvature is mentioned. For instance: "First, we assess the ability of node curvature...network characteristics (e.g. strength)." This feels somewhat strange and out of place, since the goal of this section is to discuss the different datasets used. I would recommend removing these parts.
- p10, Diffusion MRI Datasets, line 15: Here it is written "Data was acquired in...". This should at least be "Data was acquired from...", as is also done in later paragraphs. This use of "in" is also present in the next paragraph.

In addition, I would advocate to regard data as singular. This practice is adopted by several major news papers such as the Guardian and Wall Street Journal. It is up to the authors if they want to follow this advice. But please note that the Latin word agenda is also officially plural. The singular is agendum, although no one every talks about his or her agendum.
- p10, DTI Datasets from ABIDE-II: It would look better if this lonely header was on the next page.

Pim van der Hoorn

Response To Reviewers - Revision 2

Network Curvature as a Hallmark of Brain Structural Connectivity

We thank the Editor and the two Reviewers for their careful assessment of our revised manuscript. We understand that the paper was seen as significantly improved, although a few important points needed to be addressed. We provide below our point by point answer to all comments and questions, which helped further improve the paper. Significant changes to the text in the manuscript are highlighted in blue.

Reviewer 1

Reviewer's comments: In establishing (yet another) graph metric, showing that group contrasts (age, autism) identify a combination of similar and distinct brain regions does not establish that the new method establishes novel insights into structural connectivity in these applications: In particular, what do these new nodes add to our understanding of the disorder – these interpretations are rather thin and lean on reverse inference when the authors do appeal to the reported functions of the observed differences.

To address this, the authors could undertake a suitable multivariate analysis that links subject-wise structural (curvature) network changes with phenotypic differences – this approach has been well established in the community (e.g. using CCA^{1,2}) and would allow the authors to make more direct and insightful conclusions.

Response: We thank the Reviewer for raising this important question, although we respectfully note that it is not related to any of the Reviewer's initial comments. We also would like to emphasize that, although we do apply curvature-based analysis to structural brain networks in autism spectrum disorder (and other datasets), the primary goal of our manuscript is to demonstrate the utility of graph curvature broadly, and not to perform a comprehensive ASD study.

Nonetheless, we agree that the suggested analysis may help further demonstrate the strengths of our proposed method to establish novel insights into structural connectivity, and our understanding of ASD. We have therefore consulted with our colleagues from the Department of Biostatistics in order to understand what method(s) would be best suited to investigate relationship between brain network metrics and phenotypic data. We used the phenotypic data available as part of the Autism Brain Imaging Data Exchange (ABIDE) II (see <https://www.nature.com/articles/sdata201710>), from which all MRI data in our work was obtained.

We extensively experimented with canonical-correlation analysis (CCA), following the reviewer's recommendation. However, the traditional approach to CCA, or its regularized version, cannot be used in the context of our work because the number of variables (i.e., 333 nodes) is significantly larger than the number of available datasets (53 participants for SDSU, and 40 for TC data). We did attempt to perform CCA with the available phenotypic and MRI data and the node measures, only to get all correlation values of 1. As expected, this indicates that the canonical variate matrices/basis vectors are the same.

Therefore, we have performed a univariate analysis to study the relationships between nodal measures with significant differences related to ASD, and affected phenotypic measures. We present the results in Supplementary Note 5 and summarize them in the Results section of the manuscript. With the available data, univariate analysis provided useful insights into information uncovered by node curvature about ASD-related structural changes. While multivariate analysis would certainly

constitute an interesting future direction, when more MRI data becomes available in ASD, it is unfortunately challenging at this time for the reasons indicated above. Moreover, since this manuscript focuses primarily on introducing a new nodal measure for network stability and robustness characterization, we believe that such advanced multivariate analysis is beyond its scope. We hope that our efforts to relate connectivity and phenotypic data satisfy the Reviewer's question. Note that we also provide, at the end of this document, a summary of the relevant scales and scores which were found to significantly correlate with brain networks nodal measures (curvature, betweenness centrality, strength or clustering coefficient). This is only for the Reviewers' convenience and not included in Supplementary material.

Reviewer's comments: Similarly, in judging the role of curvature versus other centrality measures, it would be preferable to perform hierarchical logistic regression to understand, in direct comparison to other measures (and after accounting for some co-linearity) how much variance between the group can be accounted for by curvature.

Response: We thank the Reviewer for raising this point, which is very much in line with Comment #1 in the Analysis section of Reviewer 2's previous assessment of our manuscript ("On the other hand, if the authors really want to show that curvature is statistically different from other measures then they could also compute the correlation between these measures based on the data for each node in a single participant. This gives a single correlation value per participant and hence, using all 33 participants in the low resolution data, an empirical distribution of this correlation. If this distribution is concentrated around zero then this would be a very strong indication that curvature of a node is completely different from the other measures. This would be a great contribution to the field since it is well known that most nodal measures in the brain are correlated.").

We had addressed this point in our revision, but did not include the results into the manuscript (only in our answer to Reviewer 2), which may be the reason why Reviewer 1 did not see our answer. Since Reviewer 2 also indicated the following in their current assessment, we have now included a brief discussion on this point in sub-section "Considerations on Brain Networks Properties and Robustness Characterization" of the Methods Section, and added the Figure to the Supplementary Material. "The authors have also investigated per-node correlations between different measures as I suggested. These indeed show some correlation between curvature and the other two node-centrality measures, in particular curvature and strength. I was however surprised to find that this figure was not included in the manuscript. Since the authors argue that curvature can identify nodes that other measures cannot, these findings should, at the very least be mentioned in the discussion."

We believe that these correlation plots, albeit different from the suggested hierarchical logistic regression, provide a useful and alternative way to "judge the role of curvature versus other centrality measures". Moreover, we think that such regression methods may not be as useful for this purpose because, in order to carry out such inferential analysis in comparison to other node measures, a significantly larger sample size of MRI data would be required^{3,4}, which is beyond the scope of this paper. Our study focuses on introducing a new network nodal measure, and we perform comparative analysis to show that curvature supplements the information provided by other node measures. With currently available datasets, we find employing descriptive statistical analysis to be more appropriate, and have successfully detected brain areas with significant differences due to ASD, which are not identified by any other node measures.

Reviewer's comments: There are other recent measures of node centrality (e.g controllability², fragility⁵) that speak to the same concept as curvature: These should be incorporated into the ms. In addition, controllability – which directly engages with the stability of the system to linear perturbations – likely overlaps substantially and should be bench-marked against curvature.

Response: We thank the Reviewer for this comment, although we respectfully note again that it is not related to any of the Reviewer's initial comments. Regarding Gu *et al.*², the "controllability measure" proposed does not relate to curvature. In fact, one should note that it is an extrinsic measure, in that it requires external driving signals to actuate the network through various nodes, and quantifies their respective control authority in steering the network to various states. In contrast, curvature is intrinsic, quantifying coalescence of shortest paths between regions or return to equilibrium as a result of perturbations.

More precisely, the premise of the analysis in Gu *et al.*² is that cognitive function is driven by dynamical interactions between components of the brain. This would suggest that interaction between sites of a network are internal, possibly involving loops between neuronal pathways. Yet, the model in Gu *et al.*² assumes "external inputs" at each node/site without specifying the source of these inputs. Then, controllability reflects the ability to steer the system to equilibrium (or to any other state) and the controllability Gramian quantifies the cost of doing so. Further, it is well known in the control literature that such concepts do not directly relate to stability properties of the system or to its robustness. (Obviously, there are stable systems that are uncontrollable, and unstable systems that are controllable. Similarly for robustness to feedback stabilization, a system can be

uncontrollable with stable modes that do not affect robustness.) Additionally, we feel that the information in the supplementary material of Gu *et al.*² for their simplified linear model is not sufficient to embark on comparisons.

Regarding the reference to Gollo *et al.*⁵, one could indeed explore the fragility of hubs and the effects of cost-neutral “mutations”⁵ with the concepts introduced in our paper. However, we also feel that such an extended follow up is outside the scope of the present submission. Moreover, the publication of Gollo *et al.*⁵ is subsequent to our initial submission (February 2018), and therefore, such a follow up study would deserve a separate independent treatment.

Reviewer’s comments: One sentence reporting that statistical correction was performed with the Holm-Sidak method is insufficient: How many tests were corrected for? What is the threshold p-value? What are the other test statistics? Are the data normally distributed? This requires a brief subsection in the Methods. And a minor point is that all the p-values in the supplementary tables are specified to far too high precision (up to 8 significant positions!).

Response: We agree with the Reviewer that our initial description of the statistical methods was short. In the Methods section, we have now added a sub-section providing the requested information. We have also reformatted the supplementary tables 3 and 4 to use a consistent and shorter notation for the reported p-values.

Reviewer’s comments: It is really challenging to read and interpret the overlays of the small symbols on the surface images (Figure 4 + 5). I remain perplexed about the paragraphs in the Discussion that present new results and refer to new figures – these are anyway tangential to the main story of the paper, do not show anything about curvature, only about node strength normalization and could easily be moved to the supplementary material.

Response: Images in Figures 4 and 5 are high resolution, and we believe that they can easily be expanded to view the symbols. Moreover, we also note that Reviewer 1 felt that “The new figures for the comparison between TD and ASD brains are also much more clear and readable.” For these reasons, we decided to keep the figures in their current state.

Regarding the paragraphs in the Discussion and associated figures, a related concern was raised by Reviewer 2 (see below, second comment “The discussion in paper is more a section about using curvature for assessing network robustness. I did not find much “discussion” there at all.”). We also note that Reviewer 1 had already expressed such concern in their original review (Fourth comment: “A considerable number of the results, including all the robustness analyses, are presented in the Discussion.”) and that we did not properly address this comment. Based on Reviewer 2’s current second comment and Reviewer 1’s previous fourth comment, we respectfully disagree with Reviewer 1’s current comment about the fact that the Discussion section “do[es] not show anything about curvature”. However, we fully agree with both Reviewers that the Discussion section was not much of a “discussion”. Accordingly, we have renamed, reorganized and refocused this section, which is now incorporated into the Results Section. As requested by the Reviewer, we have also moved one figure into supplementary material, and combined the two other figures. We hope that this section now more clearly supports, in the broader context of brain networks properties, that curvature is a valuable and complementary measure to characterize graphs (brain networks) robustness.

Reviewer 2

Reviewer’s comments: The authors have also investigated per-node correlations between different measures as I suggested. These indeed show some correlation between curvature and the other two node-centrality measures, in particular curvature and strength. I was however surprised to find that this figure was not included in the manuscript. Since the authors argue that curvature can identify nodes that other measures cannot, these findings should, at the very least be mentioned in the discussion.

Response: We thank the Reviewer for the positive reception of our results on per-node correlations. We understand that this figure may indeed be useful to strengthen and clarify the message of our manuscript. We have now added the figure in Supplementary Note 8, and added a short discussion in the “Considerations on Brain Networks Properties and Robustness Characterization” sub-section of the Methods Section.

Reviewer’s comments: The discussion in paper is more a section about using curvature for assessing network robustness. I did not find much “discussion” there at all. Such a comment is also present in the report of the other referee, although the authors

do not seem to have acted upon this. I would suggest to properly rename this section and make it very clear what it is about. Here it is important to keep an eye on the storyline of this section, as in the current state the transition between the different subsections is somewhat sudden.

Response: We thank the Reviewer for raising this point, which indeed echoes Reviewer 1's previous comment ("A considerable number of the results, including all the robustness analyses, are presented in the Discussion."). Accordingly, we have renamed and reorganized this section, which is now incorporated into the Results Section (*Nature Communications* does not require a Discussion section which, if present, should anyway be succinct.) We have improved the storyline of this section to avoid the feeling of sudden transition between sub-sections.

Reviewer's comments: p1, Abstract, line (-2): Point (ii) still mentions "healthy aging" while while actually different age groups are compared. The corresponding section title has even been changed to properly reflect this. I would therefore recommend making a similar adjustment here.

Response: We thank the Reviewer for catching this. We have now removed all references to "healthy aging" and replaced them by "changes due to age", "differences between age groups", etc. as appropriate.

Reviewer's comments: p1, Introduction, line 11: "... and, thereby, that reflects..." should be "... and thereby reflects..."

Response: Corrected.

Reviewer's comments: p3, Results, line (-2): "...identify areas..." should be "...identifes areas..."

Response: Corrected.

Reviewer's comments: p4, Results, line 7: "patients perform better to detect visual targets..." should be "performs better in detecting visual targets..."

Response: Corrected.

Reviewer's comments: p4, Results, line (-5): "...literature, is known..." should be "...literature, are known..."

Response: Corrected.

Reviewer's comments: p4, Discussion, line (-8): "...betweenness centrality can...node importance than strength..." It is unclear why curvature was not included in this analysis, especially since the whole goal of the paper was to use it as a measure of node importance. Please either explain its absence or add the data to the figures.)

Response: We thank the Reviewer for identifying this issue. We have extensively re-written this section of the paper, and included curvature to the figure and analysis. We have also consolidated two of the three figures, and moved the third one into supplementary material. We note that this figure (Supplementary Note 7) does not include curvature because it is only provided as a reference to support the correctness of our analysis by reproducing the results presented in Alstott et al⁶ (Figure 3).

Reviewer's comments: p5, Conclusion, line 1: "In summary," can be removed.

Response: Corrected.

Reviewer's comments: p9, Ollivier-Ricci Curvature and Graph Robustness, line 10: I am not quite sure if "in" is the right word in "...studies his notion of curvature in the Ornstein-Uhlenbeck...". Shouldn't it be either on or for?

Response: We have replaced "in" by "for".

Reviewer's comments: p10, Measures for Brain Network Characteristics, line 3: "Suppose that the node i..." should be "Suppose that node i..."

Response: Corrected.

Reviewer's comments: p10, Diffusion MRI Datasets: Here there are three instances where the use of curvature is mentioned. For instance: "First, we assess the ability of node curvature...network characteristics (e.g. strength)." This feels somewhat strange and out of place, since the goal of this section is to discuss the different datasets used. I would recommend removing these parts.

Response: We have removed these sentences and adjusted this paragraph.

Reviewer's comments: p10, Diffusion MRI Datasets, line 15: Here it is written "Data was acquired in...". This should at least be "Data was acquired from...", as is also done in later paragraphs. This use of "in" is also present in the next paragraph.

Response: Corrected.

Reviewer's comments: In addition, I would advocate to regard data as singular. This practice is adopted by several major news papers such as the Guardian and Wall Street Journal. It is up to the authors if they want to follow this advice. But please note that the Latin word agenda is also official plural. The singular is agendum, although no one every talks about his or her agendum.

Response: Corrected.

Reviewer's comments: p10, DTI Datasets from ABIDE-II: It would look better if this lonely header was on the next page.

Response: Corrected.

References

1. Smith, S. M. *et al.* A positive-negative mode of population covariation links brain connectivity, demographics and behavior. In *Nature Neuroscience* (2015).
2. Gu, S. *et al.* Controllability of structural brain networks. In *Nature communications* (2015).
3. Peduzzi, P., Concato, J., Kemper, E., Holford, T. R. & Feinstein, A. R. A simulation study of the number of events per variable in logistic regression analysis. *J. Clin. Epidemiol.* **49**, 1373–1379 (1996).
4. Greenland, S., Schwartzbaum, J. & Finkle, W. Problems due to small samples and sparse data in conditional logistic regression analysis. *Am. journal epidemiology* **151**, 531–9 (2000).
5. Gollo, L. L. *et al.* Fragility and volatility of structural hubs in the human connectome. *Nat. Neurosci.* **21**, 1107–1116 (2018).
6. Alstott, J., Breakspear, M., Hagmann, P., Cammoun, L. & Sporns, O. Modeling the impact of lesions in the human brain. *PLOS Comput. Biol.* **5**, 1–12 (2009).

Summary of the **relevant scales and scores which were found to significantly correlate with brain networks nodal measures** curvature, betweenness centrality, strength or clustering coefficient. The full description of the ABIDE II phenotypic data legend is available at http://fcon_1000.projects.nitrc.org/indi/abide/ABIDEII_Data_Legend.pdf.

Abbreviations

ADI_R	Autism Diagnostic Interview-Revised
ADOS	Autism Diagnostic Observation Schedule
CBCL_6-18	Child Behavior Checklist Ages 6-18
DSM-IV-TR	Diagnostic and Statistical Manual of Mental Disorders Text Revision (2000)
PDD	Persistent Depressive Disorder
RBSR	Repetitive Behavior Scale-Revised
SRS	Social Responsiveness Scale

San Diego State University Dataset

Curvature

SRS_MOTIVATION_RAW	SRS Social Motivation Total Raw
SRS_MOTIVATION_T	SRS Social Motivation T Score
RBSR_6SUBSCALE_RESTRICTED	RBSR 6 Subscales Restricted Interests Subscale Raw Score
ADI_R_ONSET_TOTAL_D	Abnormality of Development Evident at or Before 36 Months
Total	
ADOS_MODULE	ADOS Module
ADOS_2_RRB	ADOS-2 Restricted and Repetitive Behavior

Betweenness Centrality

SRS_MANNERISMS_RAW	SRS Autistic Mannerisms Total Raw
SRS_MANNERISMS_T	SRS Autistic Mannerisms T Score
RBSR_6SUBSCALE_RITUALISTIC	RBSR 6 Subscales Ritualistic Behavior Subscale Raw Score
ADI_R_RRB_TOTAL_C	Restricted, Repetitive, and Stereotyped Patterns of Behavior
Total	

Strength

SRS_TOTAL_RAW	SRS Total Score
SRS_COGNITION_RAW	SRS Social Cognition T Score

Trinity College for Health Sciences

Curvature

SRS_MANNERISMS_T	SRS Autistic Mannerisms T Score
CBCL_6-18_SOCIAL_PROBLEM_T	CBCL 6-18 Social Problems T Score
CBCL_6-18_THOUGHT_T	CBCL 6-18 Thoughts Problems T Score
CBCL_6-18_ATTENTION_T	CBCL 6-18 Attention Problems T Score
CBCL_6-18_AGRESSIVE_T	CBCL 6-18 Aggressive Behavior T Score
CBCL_6-18_TOTAL_PROBLEM_T	CBCL 6-18 Total Problems T Score
RBSR_6SUBSCALE_SELF-INJURIOUS	RBSR 6 Subscales Self-Injurious Behavior Subscale Raw
Score	
RBSR_6SUBSCALE_RESTRICTED	RBSR 6 Subscales Restricted Interests Subscale Raw Score
RBSR_5SUBSCALE_SELF-INJURIOUS	RBSR 5 Subscales Self-Injurious Behavior Subscale Raw
Score	

Strength

PDD_DSM_IV_TR	Persistent Depressive Disorder DSM-IV-TR
SRS_MOTIVATION_RAW	SRS Social Motivation Total Raw
SRS_MOTIVATION_T	SRS Social Motivation T Score
CBCL_6-18_SOCIAL_PROBLEM_T	CBCL 6-18 Social Problems T Score

CBCL_6-18_THOUGHT_T
CBCL_6-18_EXTERNAL_T
CBCL_6-18_TOTAL_PROBLEM_T
ADOS_G_TOTAL
ADOS_G_SOCIAL
Interaction Score
RBSR_6SUBSCALE_RITUALISTIC
RBSR_6SUBSCALE_RESTRICTED

CBCL 6-18 Thought Problems T Score
CBCL 6-18 Externalizing Problems T Score
CBCL 6-18 Total Problems T Score
ADOS Generic or ADOS-2 Module 4 Total Score
ADOS Generic or ADOS-2 Module 4 Reciprocal Social

RBSR 6 Subscales Ritualistic Behavior Subscale Raw Score
RBSR 6 Subscales Restricted Interests Subscale Raw Score

Betweenness Centrality

PDD_DSM_IV_TR
ADI_R_SOCIAL_TOTAL_A
ADI_R_RRB_TOTAL_C

Persistent Depressive Disorder DSM-IV-TR
Reciprocal Social Interaction Sub-score Total
Restricted, Repetitive, and Stereotyped Patterns of Behavior

Total

ADOS_G_SOCIAL

ADOS Generic or ADOS-2 Module 4 Reciprocal Social

Interaction Score

RBSR_6SUBSCALE_SAMENESS

RBSR 6 Subscales Sameness Behavior Subscale Raw Score

RBSR_5SUBSCALE_TOTAL

RBSR 5 Subscales Total Raw Score

Clustering Coefficient

ADI_R_VERBAL_TOTAL_BV
ADI_R_NONVERBAL_TOTAL_BV
RBSR_6SUBSCALE_STEREOTYPED

Abnormalities in Communication Verbal Sub-score
Abnormalities in Communication Nonverbal Sub-score
RBSR 6 Subscales Stereotyped Behavior Subscale Raw

Score

RBSR_6SUBSCALE_RESTRICTED

RBSR 6 Subscales Restricted Interests Subscale Raw Score

REVIEWERS' COMMENTS:

Reviewer #1 (Remarks to the Author):

The authors have adequately addressed my prior suggestions and I endorse the paper for publication.

Yes, I focussed on the fundamental statistical flaws in the first round of reviews (lack of control for type I error), not the more nuanced issues I suggested in this round.

In terms of "incorporating" fragility and controllability into the paper, as these are related concepts, I only meant a brief discussion but can accept the push-back.

I'm pleased, despite further push-back, that the authors have chosen to remove the presentation of new results out of the Discussion section.